# Masked Autoencoders Are Effective Tokenizers for Diffusion Models

Hao Chen[*][1][2]   Yujin Han[*][3]   Fangyi Chen[1]   Xiang Li[1]   Yidong Wang[4]
Jindong Wang[5]   Ze Wang[2]   Zicheng Liu[2]   Difan Zou[3]   Bhiksha Raj[1]

## Abstract

Recent advances in latent diffusion models have demonstrated their effectiveness for high-resolution image synthesis. However, the properties of the latent space from tokenizer for better learning and generation of diffusion models remain under-explored. Theoretically and empirically, we find that improved generation quality is closely tied to the latent distributions with better structure, such as the ones with fewer Gaussian Mixture modes and more discriminative features. Motivated by these insights, we propose **MAETok**, an autoencoder (AE) leveraging mask modeling to learn semantically rich latent space while maintaining reconstruction fidelity. Extensive experiments validate our analysis, demonstrating that the variational form of autoencoders is not necessary, and a discriminative latent space from AE alone enables state-of-the-art performance on ImageNet generation using only **128** tokens. MAETok achieves significant practical improvements, enabling a gFID of **1.69** with **76×** faster training and **31×** higher inference throughput for 512×512 generation. Our findings show that the structure of the latent space, rather than variational constraints, is crucial for effective diffusion models. Code and trained models are released[1].

## 1. Introduction

Diffusion models (Sohl-Dickstein et al., 2015a; Ho et al., 2020; Rombach et al., 2022a; Peebles & Xie, 2023) have recently emerged as a powerful class of generative models, achieving state-of-the-art (SOTA) performance on various image synthesis tasks (Deng et al., 2009; Ghosh et al., 2024).

Although originally formulated in pixel space (Ho et al., 2020; Dhariwal & Nichol, 2021), subsequent research has shown that operating in a *latent space* – a compressed representation typically learned by a tokenizer – can substantially improve the efficiency and scalability of diffusion models (Rombach et al., 2022a). By avoiding the high-dimensional pixel domain during iterative diffusion and denoising steps, latent diffusion models dramatically reduce computational overhead and have quickly become the *de facto* paradigm for high-resolution generation (Esser et al., 2024).

However, a key question remains: *What constitutes a "good" latent space for diffusion*? Early work primarily employed *Variational Autoencoders* (VAE) (Kingma, 2013) as tokenizers, which ensure that the learned latent codes follow a relatively smooth distribution (Higgins et al., 2017) via a Kullback–Leibler (KL) constraint. While VAEs can empower strong generative results (Ma et al., 2024; Li et al., 2024b; Deng et al., 2024), they often struggle to achieve high pixel-level fidelity in reconstructions due to the imposed regularization (Tschannen et al., 2025). In contrast, recent explorations with *plain Autoencoders* (AE) (Hinton & Salakhutdinov, 2006; Vincent et al., 2008) produce higher-fidelity reconstructions but may yield latent spaces that are insufficiently organized or too entangled for downstream generative tasks (Chen et al., 2024b). Indeed, more recent studies emphasize that high fidelity to pixels does not necessarily translate into robust or semantically disentangled latent representations (Esser et al., 2021; Yao & Wang, 2025); leveraging latent alignment with pre-trained models can often improve generation performance further (Li et al., 2024c; Chen et al., 2024a; Qu et al., 2024; Zha et al., 2024).

In this work, we attempt to answer this question by investigating the interaction between *the latent distribution learned by tokenizers*, and *the training and sampling behavior of diffusion models* operating in that latent space. Specifically, we study AE, VAE and the recently emerging representation aligned VAE (Li et al., 2024c; Chen et al., 2024a; Zha et al., 2024; Yao & Wang, 2025), by fitting a Gaussian mixture model (GMM) into their latent space. Empirically, we show that a latent space with more *discriminative* features, whose GMM modes are *fewer*, tends to produce a lower diffusion loss. Theoretically, we prove that a latent distribution with fewer GMM modes indeed leads to a lower loss of diffusion

---

[*]Equal contribution  [1]Carnegie Mellon University  [2]AMD  [3]The University of Hong Kong  [4]Peking University  [5]William & Mary. Correspondence to: Hao Chen <haoc3@andrew.cmu.edu>.

*Proceedings of the 42$^{nd}$ International Conference on Machine Learning*, Vancouver, Canada. PMLR 267, 2025. Copyright 2025 by the author(s).

[1]https://github.com/Hhhhhhao/continuous_tokenizer.

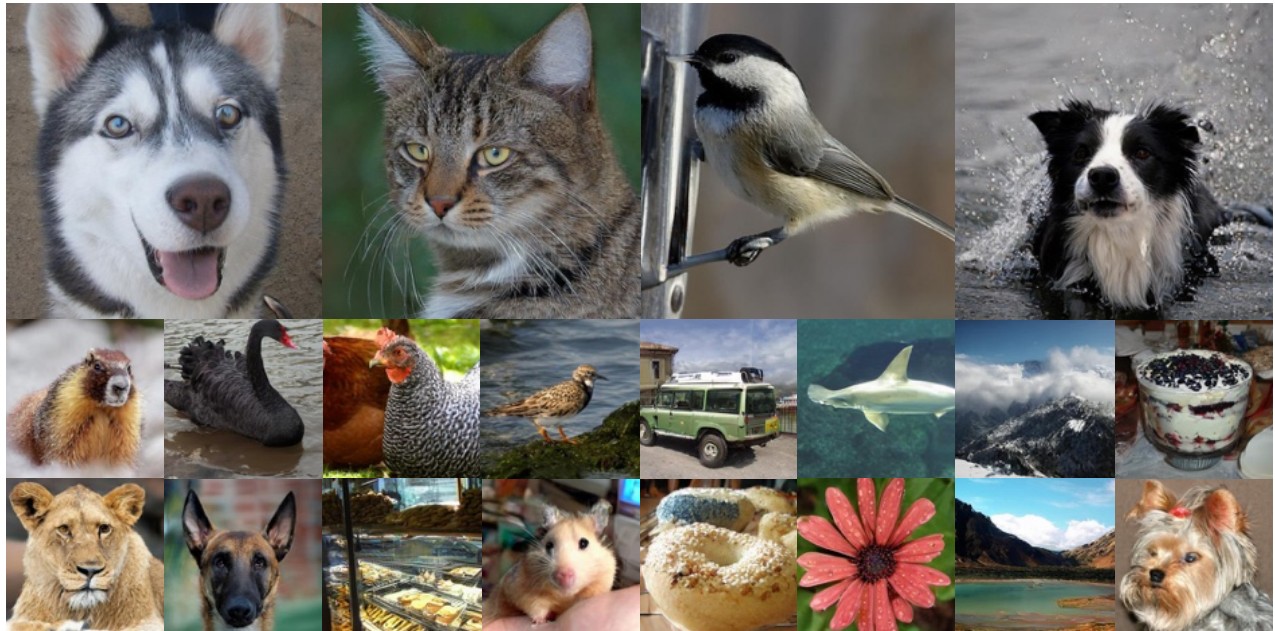

*Figure 1.* Diffusion models with MAETok achieves state-of-the-art image generation on ImageNet of 512×512 and 256×256 resolution.

models and thus to better sampling during inference.

Motivated by these insights, we demonstrate that diffusion models trained on *AE*s with discriminative latent space are enough to achieve SOTA performance. We propose to train AEs as *Masked Autoencoders* (MAE) (He et al., 2022; Xie et al., 2022; Wei et al., 2022), a self-supervised paradigm that can discover more generalized and discriminative representations by reconstructing proxy features (Zhang et al., 2022). More specifically, we adopt the transformer architecture of tokenizers (Yu et al., 2021; 2024c; Li et al., 2024c; Chen et al., 2024a) and randomly mask the image tokens at the encoder, whose features need to be reconstructed at the decoder (Assran et al., 2023). To maintain a pixel decoder with high reconstruction fidelity, we adopt auxiliary shallow decoders that predict the features of unseen tokens from seen ones to learn the representations, along with the pixel decoder which is normally trained as previous tokenizers. The auxiliary shallow decoders introduce trivial computation overhead during training. This design allows us to extend the MAE objective that reconstructs masked image patches, to simultaneously predict *multiple targets*, such as HOG (Dalal & Triggs, 2005) features (Wei et al., 2022), DINOv2 features (Oquab et al., 2023), CLIP embeddings (Radford et al., 2021; Zhai et al., 2023), and Byte-Pair Encoding (BPE) indices with text (Huang et al., 2024).

Furthermore, we reveal an interesting decoupling effect: the capacity to learn a *discriminative and semantically rich* latent space at the encoder can be separated from the capacity to *achieve high reconstruction fidelity* at the decoder. In particular, a higher mask ratio (40–60%) in MAE training often degrades immediate pixel-level quality. However, by *freez-*

*ing* the AE's encoder, thus preserving its well-organized latent space, and *fine-tuning only the decoder*, we can recover strong pixel-level reconstruction fidelity without sacrificing the semantic benefits of the learned representations.

Extensive experiments on ImageNet (Deng et al., 2009) demonstrate the effectiveness of MAETok. It addresses the trade-off between reconstruction fidelity and discriminative latent space by training the plain AEs with mask modeling, showing that the structure of latent space is more crucial for diffusion learning, instead of the variational forms of VAEs. MAETok achieves improved reconstruction FID (rFID) and generation FID (gFID) using only **128** tokens for the 256×256 and 512×512 ImageNet benchmarks.

Our contributions can be summarized as follows:

- **Theoretical and Empirical Analysis**: We establish a connection between latent space structure and diffusion model performance through both empirical and theoretical analysis. We reveal that structured latent spaces with fewer *Gaussian Mixture Model* modes enable more effective training and generation of diffusion models.

- **MAETok**: We train plain AEs using mask modeling and show that simple AEs with more discriminative latent space empower faster learning, better generation, and higher throughput of diffusion models, showing that the variational regularization of VAE is not necessary.

- **SOTA Generation Performance**: Diffusion models of 675M parameters trained on MAETok with 128 tokens achieve performance comparable to previous best models on 256 ImageNet generation and outperform previous 2B USiT at 512 resolution with 1.69 gFID and 304.2 IS.

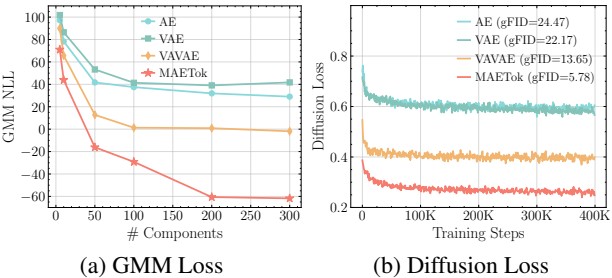

|  |  |
|---|---|
| (a) GMM Loss | (b) Diffusion Loss |

*Figure 2.* GMM fitting on latent space of AE, VAE, VAVAE, and MAETok. Fewer GMM modes in latent space usually corresponds to lower diffusion losses and better generation performance.

## 2. On the Latent Space and Diffusion Models

To study the relationship of latent space for diffusion models, we start with popular tokenizers, including AE (Hinton & Salakhutdinov, 2006), VAE (Kingma, 2013), representation aligned VAE, i.e., VAVAE (Yao & Wang, 2025). We train our own AE and VAE tokenizers under the same training recipe and the same dimension for fair comparison. We train diffusion models on them and establish connections between latent space properties and the quality of the final image generation through empirical and theoretical analysis.

**Empirical Analysis**. Inspired by existing theoretical work (Chen et al., 2022; 2023; Benton et al., 2024), our investigation of the connection between latent space and generation quality starts with a high-level intuition. With optimal diffusion model parameters, such as sufficient total time steps and adequately small discretization steps, and with assumed similar capacity of tokenizer decoders, the generation quality of diffusion models, i.e., the learned latent distribution, is dominated by the denoising network's training loss (Chen et al., 2022; 2023; Benton et al., 2024), while the effectiveness of training diffusion model via DDPM (Ho et al., 2020) heavily depends on the hardness of learning the latent space distribution (Shah et al., 2023; Diakonikolas et al., 2023; Gatmiry et al., 2024). Specially, when the training data distribution is too complex and multi-modal, i.e., not discriminative enough, the denoising network may struggle to capture such entangled global structure of latent space, resulting in a degraded generation quality.

Building upon this intuition, we use the *Gaussian Mixture Models* (GMM) to evaluate the number of modes in alternative latent space representations, where a higher number of modes indicates a more complex structure. The details of GMM training are included in Appendix B.3. Fig. 2a analyzes the GMM fitting by varying the number of Gaussian components and comparing their negative log-likelihood losses (NLL) across different latent spaces, where a lower NLL indicates better fitting quality. We observe that, to achieve comparable fitting quality, i.e., similar GMM losses, VAVAE requires fewer modes compared to VAE and AE.

Fewer modes are sufficient to adequately represent the latent space distributions of VAVAE compared to those of AE and VAE, highlighting simpler global structures in its latent space. Correspondingly, Fig. 2b reports the training losses of diffusion models with AE, VAE, and VAVAE, which (almost) align with the GMM losses shown in Fig. 2a, where fewer modes correspond to lower diffusion losses and better gFID. This alignment validates our intuition, confirming that latent spaces with fewer modes and thus more separated and discriminative features can reduce the learning difficulty and lead to better generation quality of diffusion models.

**Theoretical Analysis**. After observing experimental phenomena that align with our high-level intuition, we further present a concise theoretical analysis here to justify the rationale behind it, with more details provided in Appendix A.

Following the empirical analysis setup, we first consider a latent data distribution in $d$ dimensions modeled as a GMM with $K$ equally weighted Gaussians:

$$p_0 = \frac{1}{K} \sum_{i=1}^{K} \mathcal{N}(\boldsymbol{\mu}_i^*, \mathbf{I}), \tag{1}$$

Considering the classic diffusion model DDPM (Ho et al., 2020) and following the training objective as Shah et al. (2023), the score matching loss of DDPM at timestep $t$ is

$$\min_{\mathbf{w}} \mathbb{E}[\|s_{\mathbf{w}}(\mathbf{x}, t) - \nabla_{\mathbf{x}} \log p_t(\mathbf{x})\|^2], \tag{2}$$

where $s_{\mathbf{w}}(\mathbf{x}, t)$ represents the denoising network and $\nabla_{\mathbf{x}} \log p_t(\mathbf{x})$ denotes the oracle score function.

Then, we establish the following theorem to show that more modes typically require larger training sample sizes for diffusion models to achieve comparable generation quality.

**Theorem 2.1.** *(Informal, see Theorem A.7) Let the data distribution be a mixture of $K$ Gaussians as defined in Eq. (1). Then assume the norm of each mode is bounded by some constants, let $d$ be the data dimension, $T$ be the total time steps, and $\epsilon$ be a proper target error parameter. In order to achieve a $O(T\epsilon^2)$ error in KL divergence between data distribution and generation distirbution, the DDPM algorithm may require using $n \geq n'$ number of samples:*

$$n' = \Theta\left(\frac{K^4 d^5 B^6}{\varepsilon^2}\right), \tag{3}$$

*where the upper bound of the mean norm satisfies* $\max_i \|\boldsymbol{\mu}_i\| \leq B$.

Theorem 2.1 combines Theorem 16 from (Shah et al., 2023) and Theorem 2.2 from (Chen et al., 2023), showing that to achieve a comparable generation quality $O(T\epsilon^2)$, latent spaces with more modes ($K$) require a larger training sample size, scaling as $\mathcal{O}(K^4)$. This theoretically help explain

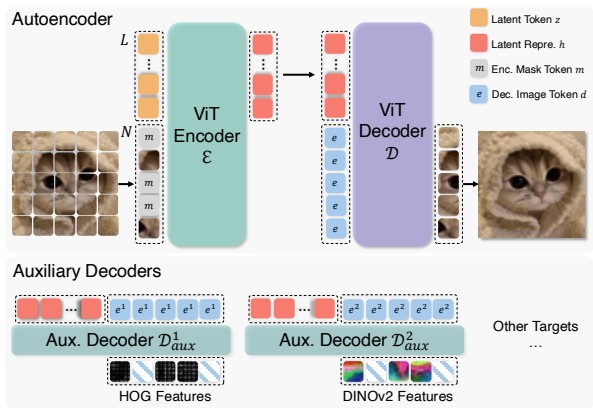

*Figure 3.* Model architecture of MAETok. We adopt the plain 1D autoencoder (AE) as tokenizer, with a vision transformer (ViT) encoder $\mathcal{E}$ and decoder $\mathcal{D}$. MAETok is trained using mask modeling at encoder, with a mask ratio of 40-60%, and predict multiple target features, e.g., HOG, DINO-v2, and CLIP features, of masked tokens from the unmasked ones using auxiliary shallow decoders.

why, under a finite number of training samples, latent spaces with more modes (e.g., AE and VAE) produce worse generations with higher gFID. We provide additional experimental results in Appendix A, demonstrating that these latent distributions share comparable upper bounds $B$, thus justifying our focus primarily on the impact of mode number $K$.

## 3. Method

Motivated by our analysis, we show that the variational form of VAEs may not be necessary for diffusion models, and simple AEs are enough to achieve SOTA generation performance with **128** tokens, as long as they have discriminative latent spaces, i.e., with fewer GMM modes. We term our method as **MAETok**, with more details as follows.

### 3.1. Architecture

We build MAETok upon the recent 1D tokenizer design with learnable latent tokens (Yu et al., 2024c; Li et al., 2024c; Chen et al., 2024a). Both the encoder $\mathcal{E}$ and decoder $\mathcal{D}$ adopt the Vision Transformer (ViT) architecture (Dosovitskiy et al., 2021; Yu et al., 2021), but are adapted to handle both image tokens and latent tokens, as shown in Fig. 3.

**Encoder**. The encoder first divides the input image $I \in \mathbb{R}^{H \times W \times 3}$ into $N$ patches according to a predefined patch size $P$, each mapped to an embedding vector of dimension $D$, resulting in image tokens $\mathbf{x} \in \mathbb{R}^{N \times D}$. In addition, we define a set of $L$ learnable latent tokens $\mathbf{z} \in \mathbb{R}^{L \times D}$. The encoder transformer takes the concatenation of image patch embeddings and latent tokens $[\mathbf{x}; \mathbf{z}] \in \mathbb{R}^{(N+L) \times D}$ as its input, and outputs the latent representations $\mathbf{h} \in \mathbb{R}^{L \times H}$ with a dimension of $H$ from only the latent tokens:

$$\mathbf{h} = \mathcal{E}\left([\mathbf{x}; \mathbf{z}]\right). \tag{4}$$

**Decoder**. To reconstruct the image, we use a set of $N$ learnable image tokens $\mathbf{e} \in \mathbb{R}^{N \times H}$. We concatenate these mask tokens with $\mathbf{h}$ as the input to the decoder, and takes only the outputs from mask tokens for reconstruction:

$$\hat{\mathbf{x}} = \mathcal{D}([\mathbf{e}; \mathbf{h}]]). \tag{5}$$

We then use a linear layer on top of $\hat{\mathbf{x}} \in \mathbb{R}^{N \times D}$ to regress the pixel values and obtain the reconstructed image $\hat{I}$.

**Position Encoding**. To encode spatial information, we apply 2D Rotary Position Embedding (RoPE) to the image patch tokens $\mathbf{x}$ at the encoder and the image tokens $\mathbf{e}$ at the decoder. In contrast, the latent tokens $\mathbf{z}$ (and their encoded counterparts $\mathbf{h}$) use standard 1D absolute position embeddings, since they do not map to specific spatial locations. This design ensures that patch-based tokens retain the notion of 2D layout, while the learned latent tokens are treated as a set of abstract features within the transformer architecture.

**Training objectives**. We train MAETok using the standard tokenizer losses as in previous work (Esser et al., 2021):

$$\mathcal{L} = \mathcal{L}_{\text{recon}} + \lambda_1 \mathcal{L}_{\text{percep}} + \lambda_2 \mathcal{L}_{\text{adv}}, \tag{6}$$

with $\mathcal{L}_{\text{recon}}$, $\mathcal{L}_{\text{percep}}$, and $\mathcal{L}_{\text{adv}}$ denoting as pixel-wise mean-square-error (MSE) loss, perceptual loss (Larsen et al., 2016; Johnson et al., 2016; Dosovitskiy & Brox, 2016; Zhang et al., 2018), and adversarial loss (Goodfellow et al., 2020; Isola et al., 2018), respectively, and $\lambda_1$ and $\lambda_2$ being hyper-parameters. Note that MAETok is a plain AE architecture, therefore, it does not require any variational loss between the posterior and prior as in VAEs, which simplifies training.

### 3.2. Mask Modeling

**Token Masking at Encoder**. A key property of MAETok is that we introduce mask modeling during training, following the principles of MAE (He et al., 2022; Xie et al., 2022), to learn a more discriminative latent space in a self-supervised way. Specifically, we randomly select a certain ratio, e.g., 40%-60%, of the image patch tokens according to a binary masking indicator $M \in \mathbb{R}^N$, and replace them with the learnable mask tokens $\mathbf{m} \in \mathbb{R}^D$ before feeding them into the encoder. *All* the latent tokens are maintained to more heavily aggregate information on the unmasked image tokens and used to reconstruct the masked tokens at the decoder output.

**Auxiliary Shallow Decoders**. In MAE, a shallow decoder (He et al., 2022) or a linear layer (Xie et al., 2022; Wei et al., 2022) is required to predict the target features, e.g., raw pixel values, HOG features, and features from pre-trained models, of the masked image tokens from the remaining ones. However, since our goal is to train MAE as tokenizers, the pixel decoder $\mathcal{D}$ needs to be able to reconstruct images in high fidelity. Thus, we keep $\mathcal{D}$ as a similar capacity to $\mathcal{E}$, and incorporate auxiliary shallow decoders to predict

additional feature targets, which share the same design as the main pixel decoder but with fewer layers. Formally, each auxiliary decoder $\mathcal{D}_{\text{aux}}^j$ takes the latent representations $\mathbf{h}$ and concatenate with their own $\mathbf{d}^j$ as inputs, and output $\hat{\mathbf{y}}^j$ as the reconstruction of their feature target $\mathbf{y}^j \in \mathbb{R}^{N \times D^j}$:

$$\hat{\mathbf{y}}^j = \mathcal{D}_{\text{aux}}^j([\mathbf{e}^j; \mathbf{h}]; \theta), \qquad (7)$$

where $D^j$ denotes the dimension of target features. We train these auxiliary decoders along with our AE using additional MSE losses at only the masked tokens according to the masking indicator $M$, similarly to Xie et al. (2022):

$$\mathcal{L}_{\text{mask}} = \sum_j \left\| M \otimes (\hat{\mathbf{y}}^j - \mathbf{y}^j) \right\|_2^2. \qquad (8)$$

### 3.3. Pixel Decoder Fine-Tuning

While mask modeling encourages the encoder to learn a better latent space, high mask ratios can degrade immediate reconstruction. To address this, after training AEs with mask modeling, we *freeze* the encoder, thus preserving the latent representations, and *fine-tune* only the pixel decoder for a small number of additional epochs. This process allows the decoder to adapt more closely to frozen latent codes of clean images, recovering the details lost during masked training. We use the same loss as in Eq. (6) for pixel decoder fine-tuning and discard all auxiliary decoders in this stage.

## 4. Experiments

We conduct comprehensive experiments to validate the design choices of MAETok, analyze its latent space, and benchmark the generation performance to show its superiority.

### 4.1. Experiments Setup

**Implementation Details of Tokenizer**. We use XQ-GAN codebase (Li et al., 2024d) to train MAETok. We use ViT-Base (Dosovitskiy et al., 2021), initialized from scratch, for both the encoder and the pixel decoder, which in total have 176M parameters. We set $L = 128$ and $H = 32$ for latent space. Three MAETok variants are trained on $256{\times}256$ ImageNet (Deng et al., 2009), and $512{\times}512$ ImageNet, and a subset of $512{\times}512$ LAION-COCO (Schuhmann et al., 2022) for 500K iterations, respectively. In the first stage training with mask modeling on ImageNet, we adopt a mask ratio of 40-60% , set by ablation, and 3 auxiliary shallow decoders for multiple targets of HOG (Dalal & Triggs, 2005), DINO-v2-Large (Oquab et al., 2023), and SigCLIP-Large (Zhai et al., 2023) features. We adopt an additional auxiliary decoder for tokenizer trained on LAION-COCO, which predicts the discrete indices of text captions for the image using a BPE tokenizer (Cherti et al., 2023; Huang et al., 2024). Each auxiliary decoder has 3 layers also set by ablation. We set $\lambda_1 = 1.0$ and $\lambda_2 = 0.4$. For the pixel decoder

fine-tuning, we linearly decrease the mask ratio from 60% to 0% over 50K iterations, with the same training loss. More training details of tokenizers are shown in Appendix B.1.

**Implementation Details of Diffusion Models**. We use SiT (Li et al., 2024a) and LightningDiT (Yao & Wang, 2025) for diffusion-based image generation tasks after training MAETok. We set the patch size of them to 1 and use a 1D position embedding, and follow their original training setting for other parameters. We use SiT-L of 458M parameters for the analysis and ablation study. For main results, we train SiT-XL of 675M parameters for 4M steps and LightningDiT for 400K steps on ImageNet of resolution 256 and 512. More details are provided in Appendix B.2.

**Evaluation**. For tokenizer evaluation, we report the reconstruction Frechet Inception Distance (rFID) (Heusel et al., 2017), peak-signal-to-noise ratio (PSNR), and structural similarity index measure (SSIM) on ImageNet and MS-COCO (Lin et al., 2014) validation set. For the latent space evaluation of the tokenizer, we conduct linear probing (LP) on the flatten latent representations and report accuracy. To evaluate the performance of generation tasks, we report generation FID (gFID), Inception Score (IS) (Salimans et al., 2016), Precision and Recall (Kynkäänniemi et al., 2019) (in Appendix C.1), with and without classifier-free guidance (CFG) (Ho & Salimans, 2022), using 250 inference steps.

### 4.2. Design Choices of MAETok

We first present an extensive ablation study to understand how mask modeling and different designs affect the reconstruction of tokenizer and, more importantly, the generation of diffusion models. We start with an AE and add different components to study both rFID of AE and gFID of SiT-L.

**Mask Modeling**. In Table 1a, we compare AE and VAE with mask modeling and also study the proposed fine-tuning of the pixel decoder. For AE, mask modeling significantly improves gFID and slightly deteriorates rFID, which can be recovered through the decoder fine-tuning stage without sacrificing generation performance. In contrast, mask modeling only marginally improves the gFID of VAE, since the imposed KL constraint may hinder latent space learning.

**Reconstruction Target**. In Table 1b, we study how different reconstruction targets affect latent space learning in mask modeling. We show that using the low-level reconstruction features, such as the raw pixel (with only a pixel decoder) and HOG features, can already learn a better latent space, resulting in a lower gFID. Adopting semantic teachers such as DINO-v2 and CLIP instead can significantly improve gFID. Combining different reconstruction targets can achieve a balance in reconstruction fidelity and generation quality.

**Mask Ratio**. In Table 1c, we show the importance of proper mask ratio for learning the latent space using HOG target,

| case | rFID | gFID |
|------|------|------|
| VAE | 1.22 | 22.17 |
| +MM | 1.75 | 18.17 |
| AE | 0.67 | 24.47 |
| +MM | 0.85 | 5.78 |
| +FT | 0.48 | 5.69 |

(a) Mask modeling.

| case | rFID | gFID |
|------|------|------|
| pixel | 1.15 | 17.18 |
| HOG | 2.43 | 13.54 |
| DINO | 0.89 | 6.24 |
| CLIP | 0.78 | 11.31 |
| Comb. | 0.85 | 5.78 |

(b) Reconstruction target.

| low | high | rFID | gFID |
|-----|------|------|------|
| 0 | 60 | 0.82 | 24.15 |
| 10 | 40 | 1.01 | 22.63 |
| 20 | 60 | 1.44 | 20.35 |
| 40 | 40 | 1.78 | 18.27 |
| 40 | 60 | 2.43 | 17.18 |

(c) Mask ratio (HOG w/o FT).

| blocks | rFID | gFID |
|--------|------|------|
| linear | 1.35 | 6.98 |
| 1 | 1.19 | 6.43 |
| 3 | 0.85 | 5.78 |
| 6 | 0.86 | 7.12 |
| 12 | 0.96 | 8.80 |

(d) Aux. decoder depth.

*Table 1.* Ablations with MAETok on 256×256 ImageNet. We report rFID of tokenizer and gFID of SiT-L trained on latent space of the tokenizer without classifier-free guidance. We train tokenizer of 250K and SiT-L for 400K steps. Default settings are indicated in  Grey .

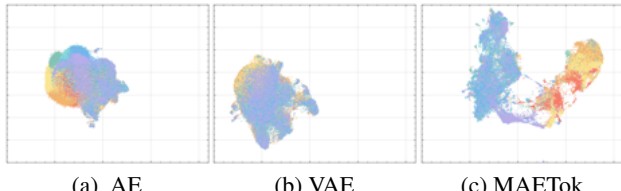

(a) AE  (b) VAE  (c) MAETok

*Figure 4.* UMAP visualization on ImageNet of the learned latent space from (a) AE; (b) VAE; (c) MAETok. Colors indicate different classes. MAETok presents a more discriminative latent space.

as highlighted in previous works (He et al., 2022; Wei et al., 2022; Xie et al., 2022). A low mask ratio prevents the AE from learning more discriminative latent space. A high mask ratio imposes a trade-off between reconstruction fidelity and the latent space quality, and thus generation performance.

**Auxiliary Decoder Depth**. We study the depth of auxiliary decoder in Table 1d with multiple reconstruction targets. We show that a decoder that is too shallow or too deep could hurt both the reconstruction fidelity and generation quality. When the decoder is too shallow, the combined target features may confuse the latent with high-level semantics and low-level details, resulting in a worse reconstruction fidelity. However, a deeper auxiliary decoder may learn a less discriminative latent space of the AE with its strong capacity, and thus also lead to worse generation performance.

We include more ablation study on the number of learnable latent tokens and 2D RoPE in Appendix C.4.

### 4.3. Latent Space Analysis

We further analyze the relationship between the latent space of the AE variants and the generation performance of SiT-L.

**Latent Space Visualization**. We provide a UMAP visualization (McInnes et al., 2018) in Fig. 4 to intuitively compare the latent space learned by different variants of AE. Notably, both the AE and VAE exhibit more entangled latent embeddings, where samples corresponding to different classes tend to overlap substantially. In contrast, MAETok shows distinctly separated clusters with relatively clear boundaries between classes, suggesting that MAETok learns more discriminative latent representations. In line with our analysis in Section 2 and Fig. 2, a more discrimina-

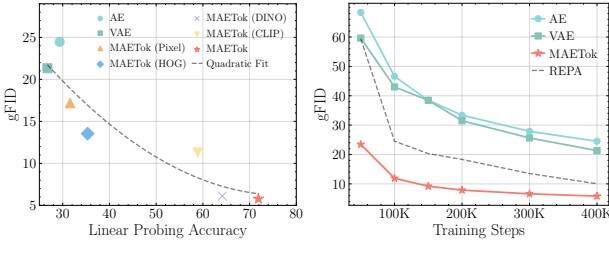

(a) gFID vs. LP Acc.  (b) gFID during training

*Figure 5.* The latent space from tokenizer correlates strongly with generation performance. More discriminative latent space (a) with higher linear probing (LP) accuracy usually leads to better gFID, and (b) makes the learning of the diffusion model easier and faster.

tive and separated latent representation of MAETok results in much fewer GMM modes and improve the generation performance. More visualization is shown in Appendix C.3.

**Latent Distribution and Generation Performance**. We assess the latent space's quality by studying the relationship between the linear probing (LP) accuracy on the latent space, as a proxy of how well semantic information is preserved in the latent codes, and the gFID for generation performance. In Fig. 5a, we observe tokenizers with more discriminative latent distributions, as indicated by higher LP accuracy, correspondingly achieve lower gFID. This finding suggests that when features are well-clustered in latent space, the generator can more easily learn to generate high-fidelity samples. We further verify this intuition by tracking gFID throughout training, shown in Fig. 5b, where MAETok enables faster convergence, with gFID rapidly decreasing with lower values than the AE or VAE baselines. A high-quality latent distribution is shown to be a crucial factor in both achieving strong final generation metrics and accelerating training.

### 4.4. Main Results

**Generation**. We compare SiT-XL and LightningDiT based on variants of MAETok in Tables 2 and 3 for the 256×256 and 512×512 ImageNet benchmarks, respectively, against other SOTA generative models. Notably, the **naive SiT-XL trained on MAETok with only 128 tokens and plain AE architecture** achieves consistently better gFID and IS without using CFG: it outperforms REPA (Yu et al., 2024d) by **3.59** gFID on 256 resolution and establishes a SOTA com-

| Model (G) | # Params (G) | Model (T) | # Params (T) | # Tokens ↓ | rFID ↓ | w/o CFG gFID ↓ | IS ↑ | w/ CFG gFID ↓ | IS ↑ |
|---|---|---|---|---|---|---|---|---|---|
| *Auto-regressive* | | | | | | | | | |
| VQGAN (Esser et al., 2021) | 1.4B | VQ | 23M | 256 | 7.94 | – | – | 5.20 | 290.3 |
| ViT-VQGAN (Yu et al., 2021) | 1.7B | VQ | 64M | 1024 | 1.28 | 4.17 | 175.1 | – | – |
| RQ-Trans. (Lee et al., 2022) | 3.8B | RQ | 66M | 256 | 3.20 | – | – | 3.80 | 323.7 |
| MaskGIT (Chang et al., 2022) | 227M | VQ | 66M | 256 | 2.28 | 6.18 | 182.1 | – | – |
| LlamaGen-3B (Sun et al., 2024) | 3.1B | VQ | 72M | 576 | 2.19 | – | – | 2.18 | 263.3 |
| TiTok-S-128 (Yu et al., 2024c) | 287M | VQ | 72M | 128 | 1.61 | – | – | 1.97 | 281.8 |
| VAR (Tian et al., 2024) | 2B | MSRQ† | 109M | 680 | 0.90 | – | – | 1.92 | 323.1 |
| ImageFolder (Li et al., 2024c) | 362M | MSRQ | 176M | 286 | 0.80 | – | – | 2.60 | 295.0 |
| MAGVIT-v2 (Yu et al., 2024a) | 307M | LFQ | 116M | 256 | 1.61 | 3.07 | 213.1 | 1.78 | 319.4 |
| MaskBit (Weber et al., 2024) | 305M | LFQ | 54M | 256 | 1.61 | – | – | 1.52 | 328.6 |
| MAR-H (Li et al., 2024b) | 943M | KL | 66M | 256 | 1.22 | 2.35 | 227.8 | 1.55 | 303.7 |
| *Diffusion-based* | | | | | | | | | |
| LDM-4 (Rombach et al., 2022b) | 400M | KL† | 55M | 4096 | 0.27 | 10.56 | 103.5 | 3.60 | 247.7 |
| U-ViT-H/2 (Bao et al., 2023) | 501M | | | | | – | – | 2.29 | 263.9 |
| MDTv2-XL/2 (Gao et al., 2023) | 676M | | | | | 5.06 | 155.6 | 1.58 | 314.7 |
| DiT-XL/2 (Peebles & Xie, 2023) | 675M | KL† | 84M | 1024 | 0.62 | 9.62 | 121.5 | 2.27 | 278.2 |
| SiT-XL/2 (Ma et al., 2024) | 675M | | | | | 8.30 | 131.7 | 2.06 | 270.3 |
| + REPA (Yu et al., 2024d) | 675M | | | | | 5.90 | 157.8 | 1.42 | 305.7 |
| TexTok-256 (Zha et al., 2024) | 675M | KL | 176M | 256 | 0.69 | – | – | 1.46 | 303.1 |
| LightningDiT (Yao & Wang, 2025) | 675M | KL | 70M | 256 | 0.28 | 2.17 | 205.6 | 1.35 | 295.3 |
| *Ours* | | | | | | | | | |
| MAETok + LightningDiT | 675M | AE | 176M | **128** | 0.48 | 2.21 | 208.3 | 1.73 | 308.4 |
| MAETok + SiT-XL | 675M | AE | 176M | **128** | 0.48 | 2.31 | 216.5 | 1.67 | 311.2 |

*Table 2.* System-level comparison on ImageNet 256×256 conditional generation. SiT-XL and LightningDiT trained on MAETok achieves performance comparable to state-of-the-art using plain AE with only 128 tokens. "Model (G)": the generation model. "# Params (G)": the number of generator's parameters. "Model (T)": the tokenizer model. "# Params (T)": the number of tokenizer's parameters. "# Tokens": the number of latent tokens used during generation. † indicates that the model has been trained on other data than ImageNet.

| Model (G) | # Params (G) | Model (T) | # Params (T) | # Tokens ↓ | rFID ↓ | w/o CFG gFID ↓ | IS ↑ | w/ CFG gFID ↓ | IS ↑ |
|---|---|---|---|---|---|---|---|---|---|
| *GAN* | | | | | | | | | |
| BigGAN (Chang et al., 2022) | – | – | – | – | – | – | – | 8.43 | 177.9 |
| StyleGAN-XL (Karras et al., 2019) | 168M | – | – | – | – | – | – | 2.41 | 267.7 |
| *Auto-regressive* | | | | | | | | | |
| MaskGIT (Chang et al., 2022) | 227M | VQ | 66M | 1024 | 1.97 | 7.32 | 156.0 | – | – |
| TiTok-B-64 (Yu et al., 2024c) | 177M | VQ | 202M | 128 | 1.52 | – | – | 2.13 | 261.2 |
| MAGVIT-v2 (Yu et al., 2024a) | 307M | LFQ | 116M | 1024 | - | – | – | 1.91 | 324.3 |
| MAR-H (Li et al., 2024b) | 943M | KL | 66M | 1024 | – | 2.74 | 205.2 | 1.73 | 279.9 |
| *Diffusion-based* | | | | | | | | | |
| ADM (Dhariwal & Nichol, 2021) | – | – | – | – | – | 23.24 | 58.06 | 3.85 | 221.7 |
| U-ViT-H/4 (Bao et al., 2023) | 501M | | | | | – | – | 4.05 | 263.8 |
| DiT-XL/2 (Peebles & Xie, 2023) | 675M | KL† | 84M | 4096 | 0.62 | 9.62 | 121.5 | 3.04 | 240.8 |
| SiT-XL/2 (Ma et al., 2024) | 675M | | | | | – | – | 2.62 | 252.2 |
| DiT-XL (Chen et al., 2024b) | 675M | | | | | 9.56 | – | 2.84 | – |
| UViT-H (Chen et al., 2024b) | 501M | | | | | 9.83 | – | 2.53 | – |
| UViT-H (Chen et al., 2024b) | 501M | AE† | 323M | 256 | 0.22 | 12.26 | – | 2.66 | – |
| UViT-2B (Chen et al., 2024b) | 2B | | | | | 6.50 | – | 2.25 | – |
| USiT-2B (Chen et al., 2024b) | 2B | | | | | 2.90 | – | 1.72 | – |
| *Ours* | | | | | | | | | |
| MAETok + LightningDiT | 675M | | | | | 2.56 | 224.5 | 1.72 | 307.3 |
| MAETok + SiT-XL | 675M | AE | 176M | **128** | 0.62 | 2.79 | 204.3 | **1.69** | 304.2 |
| MAETok + USiT-2B | 2B | | | | | **1.72** | **244.3** | **1.65** | **312.5** |

*Table 3.* System-level comparison on ImageNet 512×512 conditional generation. SiT-XL and LightningDiT trained on MAETok achieve state-of-the-art performance using plain AE with only 128 tokens, outperforming USiT of 2B parameters using only 675M parameters.

parable gFID of **2.79** at 512 resolution. When using CFG, SiT-XL achieves a comparable performance with competing autoregressive and diffusion-based baselines trained on VAEs at 256 resolution. It beats the 2B USiT (Chen et al., 2024b) with 256 tokens and also achieves a new SOTA of **1.69** gFID and **304.2** IS at 512 resolution. Better results

have been observed with LightningDiT, trained with more advanced tricks (Yao & Wang, 2025), where it outperforms MAR-H of 1B parameters and USiT of 2B parameters without CFG, achieves a **2.56** gFID and **224.5** IS, and **1.72** gFID with CFG. When using a USiT-2B (Chen et al., 2024b) for 512 generation, it pushes the gFID without CFG to **1.72**,

| Tokenizer | # Params | # Tokens | ImageNet | | | COCO | | |
|---|---|---|---|---|---|---|---|---|
| | | | rFID↓ | PSNR↑ | SSIM↑ | rFID↓ | PSNR↑ | SSIM↑ |
| *256×256* | | | | | | | | |
| SD-VAE[†] | 84M | 1024 | 0.62 | 26.04 | 0.834 | 4.07 | 25.76 | 0.845 |
| DC-AE[†] | 323M | 64 | 0.77 | 23.93 | 0.766 | 5.10 | 23.59 | 0.776 |
| VA-VAE | 70M | 256 | 0.28 | 26.30 | 0.846 | 2.80 | 26.12 | 0.856 |
| SoftVQ | 176M | 64 | 0.61 | 22.97 | 0.739 | 5.16 | 22.86 | 0.745 |
| TexTok | 176M | 256 | 0.69 | 24.38 | 0.645 | - | - | - |
| MAETok | 176M | 128 | 0.48 | 23.61 | 0.763 | 4.87 | 23.31 | 0.773 |
| *512×512* | | | | | | | | |
| SD-VAE[†] | 84M | 4096 | 0.19 | 27.36 | 0.849 | 2.41 | 26.48 | 0.841 |
| DC-AE[†] | 323M | 256 | 0.21 | 26.23 | 0.815 | 2.85 | 25.47 | 0.811 |
| TexTok | 176M | 256 | 0.73 | 24.45 | 0.668 | - | - | - |
| MAETok | 176M | 128 | 0.62 | 22.18 | 0.701 | 5.91 | 22.48 | 0.695 |
| MAETok[†] | 176M | 128 | 0.76 | 22.43 | 0.717 | 5.25 | 23.35 | 0.684 |

*Table 4.* Comparison of various continuous tokenizers. † indicates the tokenizer is trained on other data than ImageNet. MAETok achieves a better trade-off of compression and reconstruction.

and gFID with CFG to **1.65**. These results demonstrate that **the structure of the latent space** (see Fig. 4), instead of the variational form of tokenizers, is vital for the diffusion model to learn effectively and efficiently. We show a few selected generation samples in Fig. 1, and more uncurated visualizations are included in Appendix C.5.

**Reconstruction**. MAETok also offers strong reconstruction capabilities on ImageNet and MS-COCO, as shown in Table 4. Compared to previous continuous tokenizers, including SD-VAE (Rombach et al., 2022a), DC-AE (Chen et al., 2024b), VA-VAE (Yao & Wang, 2025), SoftVQ-VAE (Chen et al., 2024a), and TexTok (Zha et al., 2024), MAE-Tok achieves a favorable trade-off between the quality of the reconstruction and the size of the latent space. On 256×256 ImageNet, using **128 tokens**, MAETok attains an rFID of **0.48** and SSIM of **0.763**, outperforming methods such as SoftVQ in terms of both fidelity and perceptual similarity, while using half of the tokens in TexTok (Zha et al., 2024). On MS-COCO, where the tokenizer is not directly trained, MAETok still delivers robust reconstructions. At resolution of 512, MAETok maintains its advantage by balancing compression ratio and the reconstruction quality.

### 4.5. Discussion

**Efficient Training and Generation**. A prominent benefit of the 1D tokenizer design is that it enables arbitrary number of latent tokens. The 256×256 and 512×512 images are usually encoded to 256 and 1024 tokens, while MAETok uses **128** tokens for both. It allows for much more efficient training and inference of diffusion models. For example, when using 1024 tokens of 512×512 images, the Gflops and the inference throughput of SiT-XL are 373.3 and 0.1 images/second on a single A100, respectively. MAETok reduces the Glops to **48.5** and increases throughput to **3.12** images/second. With improved convergence, MAETok enables a **76x** faster training to perform similarly to REPA.

**Unconditional Generation**. An interesting observation

from our results is that diffusion models trained on MAETok usually present significantly better generation performance without CFG, compared to previous methods, yet smaller performance gap with CFG. We hypothesize that the reason is that the unconditional class also learns the semantics in the latent space, as shown by the unconditional generation

| Metric | AE | VAE | MAETok (HOG) | MAETok (CLIP) | MAETok (DINO) | MAETok |
|---|---|---|---|---|---|---|
| gFID | 59.02 | 58.34 | 45.31 | 34.73 | 20.76 | 18.31 |
| IS | 16.91 | 17.36 | 24.25 | 28.33 | 44.51 | 47.33 |

*Table 5.* Unconditional generation performance of SiT-L.

performance in Table 5. As the latent space becomes more discriminative, the unconditional generation performance also improves significantly. This implies that the CFG linear combination scheme may become less effective (Zhao & Schwing, 2025), aligning with our CFG tuning results included in Appendix C.2. Moreover, adopting more recent advanced CFG techniques, such as Autoguidance (Karras et al., 2024) with naively earlier checkpoints of the generative model, and guidance-free training (Chen et al., 2025) can further improve the gFID of the SiT-XL model from 1.67 to 1.54 and 1.51, respectively. We leave the exploration of other CFG techniques for future work.

**Learnable Tokens, Mask Modeling, and Auxiliary Decoders**. We also provide a study of the effect of each component in MAETok, as shown in Table 6. The results reveal clear and complementary gains from each design choice. When all three are present, MAETok reaches a better trade-off between the reconstruction and generation quality; dropping the auxiliary decoder or the masking objective, for instance, leads to noticeably weaker results, while omitting the learnable tokens, i.e., using only 256 image tokens, impairs both fidelity metrics most severely.

| Mask Modeling | Learnable Token | Aux. Decoder | rFID | gFID |
|---|---|---|---|---|
| ✓ | ✓ | ✓ | 0.85 | 5.78 |
| | ✓ | ✓ | 0.64 | 8.44 |
| ✓ | | ✓ | 1.01 | 6.85 |
| ✓ | ✓ | | 1.15 | 17.18 |
| | | ✓ | 0.43 | 9.88 |
| ✓ | | | 0.96 | 18.23 |
| | ✓ | | 0.67 | 24.47 |

*Table 6.* Effects of different components in MAETok.

## 5. Related Work

**Image Tokenization**. Imgae tokenization aims at transforming the high-dimension images into more compact and structured latent representations. Early explorations mainly used autoencoders (Hinton & Salakhutdinov, 2006; Vincent et al., 2008), which learn latent codes reduced dimensionality. These foundations soon inspired methods with variational posteriors, such as VAEs (Van Den Oord et al.,

2017; Razavi et al., 2019a) and VQ-GAN (Esser et al., 2021; Razavi et al., 2019b). Recent work has further improved compression fidelity and scalability (Lee et al., 2022; Yu et al., 2024a; Mentzer et al., 2023; Zhu et al., 2024), showing the importance of latent structure. More recent efforts have shown methods that bridge high-fidelity reconstruction and semantic understanding within a single tokenizer (Yu et al., 2024c; Li et al., 2024c; Chen et al., 2024a; Wu et al., 2024; Gu et al., 2023). Complementary to them, we further highlight the importance of discriminative latent space, which allows us to use a simple AE yet achieve better generation.

**Image Generation**. The paradigms of image generation mainly categorize to autoregressive and diffusion models. Autoregressive models initially relied on CNN architectures (Van den Oord et al., 2016) and were later augmented with Transformer-based models (Vaswani et al., 2023; Yu et al., 2024b; Lee et al., 2022; Liu et al., 2024; Sun et al., 2024) for improved scalability (Chang et al., 2022; Tian et al., 2024). Diffusion models show strong performance since their debut Sohl-Dickstein et al. (2015b). Key developments (Nichol & Dhariwal, 2021; Dhariwal & Nichol, 2021; Song et al., 2022) refined the denoising process for sharper samples. A pivotal step in performance and efficiency came with latent diffusion (Vahdat et al., 2021; Rombach et al., 2022b), which uses tokenizers to reduce dimension and conduct denoising in a compact latent space (Van Den Oord et al., 2017; Esser et al., 2021; Peebles & Xie, 2023; Qiu et al., 2025). Recent advances include designing better tokenizers (Chen et al., 2024a; Zha et al., 2024; Yao & Wang, 2025) and combining diffusion with autoregressive models (Li et al., 2024b).

## 6. Conclusion

We presented a theoretical and empirical analysis of latent space properties for diffusion models, demonstrating that fewer modes in latent distributions enable more effective learning and better generation quality. Based on these insights, we developed MAETok, which achieves state-of-the-art performance through mask modeling without requiring variational constraints. Using only 128 tokens, our approach significantly improves both computational efficiency and generation quality on ImageNet. Our findings establish that a more discriminative latent space, rather than variational constraints, is crucial for effective diffusion models, opening new directions for efficient generative modeling at scale.

## Impact Statement

This work advances the fundamental understanding and technical capabilities of machine learning systems, specifically in the domain of image generation through diffusion models. While our contributions are primarily technical, improving efficiency and effectiveness of generative models, we acknowledge that advances in image synthesis technology can have broader societal implications. These may include both beneficial applications in creative tools and design, as well as potential concerns regarding synthetic media. We have focused on developing more efficient and robust methods for image generation, and we encourage ongoing discussion about the responsible deployment of such technologies.

## Acknowledge

The authors would like to thank the anonymous reviewers and area chair for their helpful comments. This research project has benefited from the Microsoft Accelerating Foundation Models Research (AFMR) grant program. Difan Zou and Yujin Han acknowledge the support from NSFC 62306252, Hong Kong ECS award 27309624, Guangdong NSF 2024A1515012444, and the central fund from HKU.

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

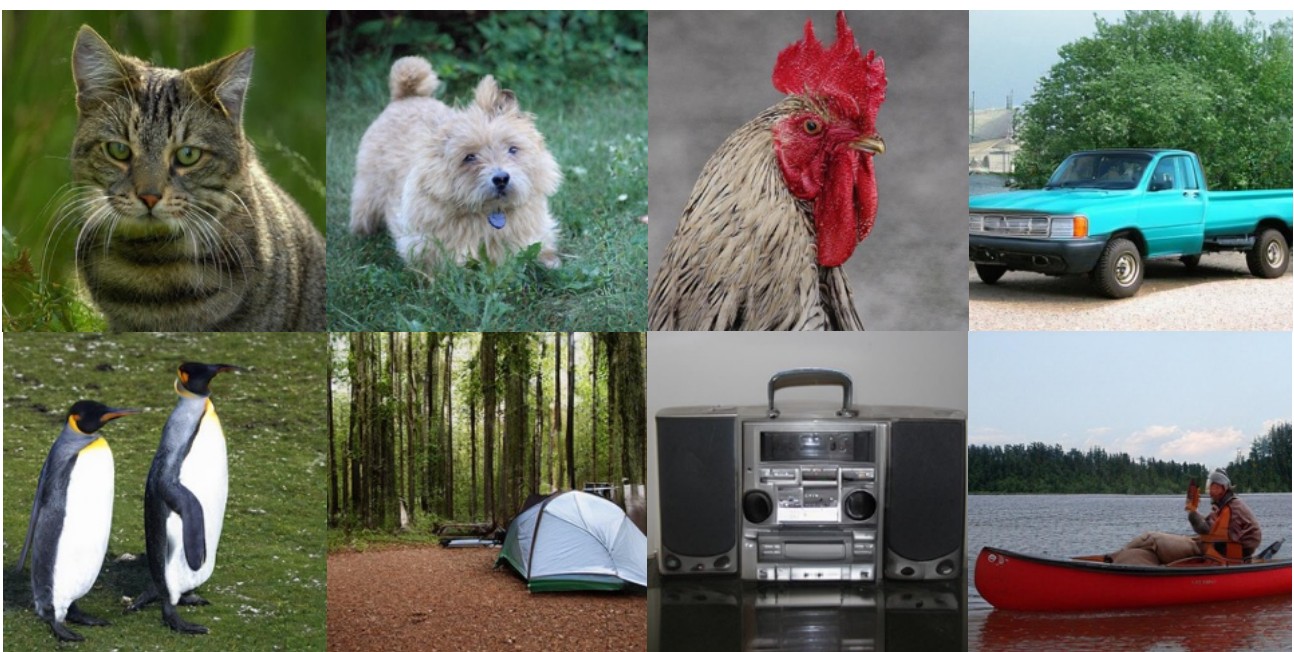

*Figure 6.* Additional selected samples from $512{\times}512$ SiT-XL model on MAETok. We use a classifier-free guidance scale of 2.0.

## A. Theoretical Analysis

**Preliminary.** We begin the theoretical analysis by introducing the preliminaries of the problem and the necessary notation.

Following the empirical analysis setting, we first consider the latent data distribution is the GMM with $K$ equally weighted Gaussians:

$$p_0 = \frac{1}{K} \sum_{i=1}^{K} \mathcal{N}(\boldsymbol{\mu}_i^*, \mathbf{I}), \tag{9}$$

Following the the training objective (Shah et al., 2023), we consider the score matching loss of DDPM at timestep $t$ is

$$\min_{\mathbf{w}} \mathbb{E}[\|s_{\mathbf{w}}(\mathbf{x}, t) - \nabla_{\mathbf{x}} \log p_t(\mathbf{x})\|^2] \tag{10}$$

where $s_{\mathbf{w}}(\mathbf{x}, t)$ is the denoising network and $\log p_t(\mathbf{x})$ is the oracle score. Under the GMM assumption, the explicit solution of score function $\nabla_{\mathbf{x}} \log p_t(\mathbf{x})$ can be written as

$$\nabla_{\mathbf{x}} \ln p_t(\mathbf{x}) = \sum_{i=1}^{K} w_{i,t}^*(\mathbf{x}) \boldsymbol{\mu}_{i,t}^* - \mathbf{x}, \tag{11}$$

where the weighting parameter is

$$w_{i,t}^*(\mathbf{x}) := \frac{\exp(-\|\mathbf{x} - \boldsymbol{\mu}_{i,t}^*\|^2/2)}{\sum_{j=1}^{K} \exp(-\|\mathbf{x} - \boldsymbol{\mu}_{j,t}^*\|^2/2)}, \quad \boldsymbol{\mu}_{i,t}^* := \boldsymbol{\mu}_i^* \exp(-t). \tag{12}$$

Therefore, we can consider the denosing neural network with the following format, that is

$$s_{\theta_t}(\mathbf{x}) = \sum_{i=1}^{K} w_{i,t}(\mathbf{x}) \boldsymbol{\mu}_{i,t} - \mathbf{x}, \tag{13}$$

where

$$w_{i,t}(\mathbf{x}) := \frac{\exp(-\|\mathbf{x} - \boldsymbol{\mu}_{i,t}\|^2/2)}{\sum_{j=1}^{K} \exp(-\|\mathbf{x} - \boldsymbol{\mu}_{j,t}\|^2/2)}, \quad \boldsymbol{\mu}_{i,t} := \boldsymbol{\mu}_i \exp(-t). \tag{14}$$

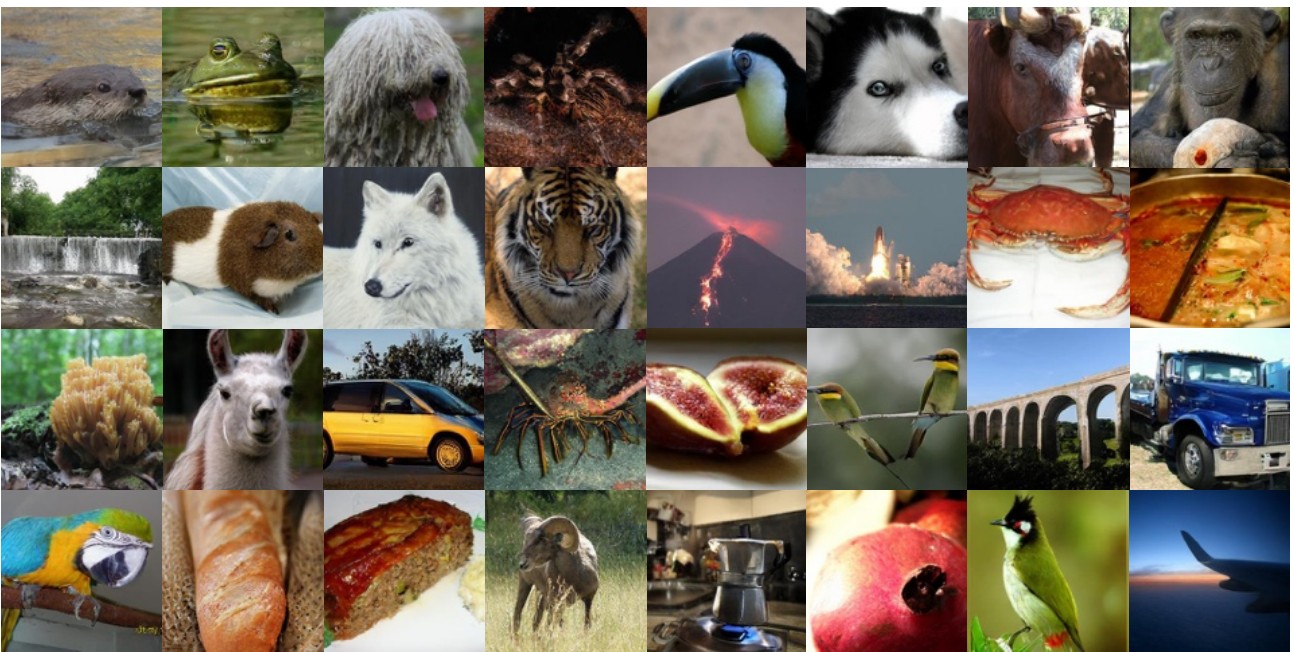

*Figure 7.* Additional selected samples from $256\times256$ diffusion models on MAETok. We use a classifier-free guidance scale of 2.0.

**Assumptions.** To ensure the denoising network approximates the score function with sufficient accuracy, we consider the following three common assumptions, which constrain the training process from the perspectives of data quality (separability), good initialization (warm start), and regularity (bounded mean of target distribution) (Chen et al., 2022; 2023; Benton et al., 2024).

**Assumption A.1.** (Separation Assumption in (Shah et al., 2023)) For a mixture of $K$ Gaussians given by Equation 9, for every pair of components $i, j \in \{1, 2, \ldots, K\}$ with $i \neq j$, we assume that the separation between their means

$$\|\boldsymbol{\mu}_i^* - \boldsymbol{\mu}_j^*\| \geq C\sqrt{\log(\min(K, d))} \tag{15}$$

for sufficiently large absolute constant $C > 0$.

**Assumption A.2.** (Initialization Assumption in (Shah et al., 2023)) For each component $i \in \{1, 2, \ldots, K\}$, we have an initialization $\boldsymbol{\mu}_i^{(0)}$ with the property that

$$\|\boldsymbol{\mu}_i^{(0)} - \boldsymbol{\mu}_i^*\| \leq C'\sqrt{\log(\min(K, d))} \tag{16}$$

for sufficiently small absolute constant $C' > 0$.

**Assumption A.3.** The maximum mean norm of the GMM in GMM 9 is bounded as: $\max_i \|\boldsymbol{\mu}_i\| \leq B$.

*Remark* A.4. By Assumption A.3, we could derive the second movement bound of $p_0$ as

$$\mathbb{E}_{\mathbf{x}\sim p_0}[\|\mathbf{x}\|^2] = \int p_0(\mathbf{x})\|\mathbf{x}\|^2 \mathrm{d}\mathbf{x} \leq d + B^2 \tag{17}$$

Then, we can have the following analysis,

**Step 1: From $K$ Modes to Training Loss.** The main conclusion required for our proof is derived from the following theorem, which provides the estimation error $\|\boldsymbol{\mu}_i - \boldsymbol{\mu}_i^*\|$ for DDPM with gradient descent under $\mathcal{O}(1)$-level noise, assuming that Assumptions A.1 and A.2 are satisfied.

**Theorem A.5.** *(Adopted from Theorem 16 in Shah et al. (2023)) Let q be a mixture of Gaussians in Eq. (9) with center parameters $\theta^* = \{\boldsymbol{\mu}_1^*, \boldsymbol{\mu}_2^*, \ldots, \boldsymbol{\mu}_K^*\} \in \mathbb{R}^d$ satisfying the separation A.1, and suppose we have estimates $\theta$ for the centers*

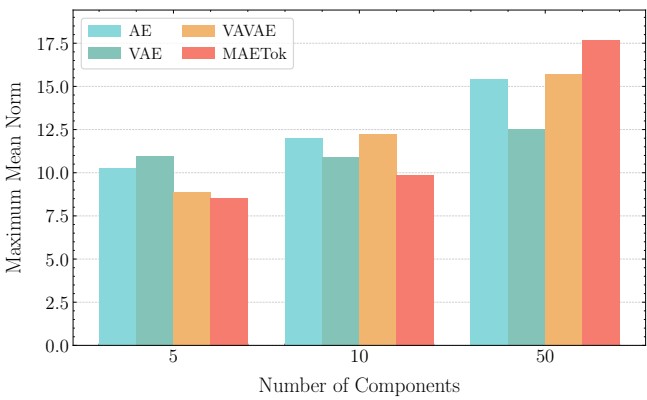

*Figure 8.* We compare the maximum mean norm across different numbers of components and observe that AE, VAE, VAVAE, and our method MAETok exhibit similar maximum mean norms. This suggests that these latent spaces share a comparable prior upper bound $B$, supporting the rationale for primarily considering the number of modes, i.e., $K$ in Theorem 2.1.

*such that the warm initialization Assumption A.2 is satisfied. For any $\varepsilon > \varepsilon_0$ and noise scale $t$ where*

$$\varepsilon_0 = 1/poly(d) \quad t = \Theta(\varepsilon),$$

*gradient descent on the DDPM objective at noise scale $t$ outputs $\tilde{\theta} = \{\tilde{\boldsymbol{\mu}}_1, \tilde{\boldsymbol{\mu}}_2, \ldots, \tilde{\boldsymbol{\mu}}_K\}$ such that $\min_i \|\tilde{\boldsymbol{\mu}}_i - \boldsymbol{\mu}_i^*\| \leq \varepsilon$ with high probability. DDPM runs for $H \geq H'$ iterations and uses $n \geq n'$ number of samples where*

$$H' = \Theta(\log(\varepsilon^{-1} \log d)), \quad n' = \Theta(K^4 d^5 B^6 / \varepsilon^2).$$

Theorem A.5 indicates that, to achieve the same estimation error $\epsilon$, a data distribution with more modes requires more training samples. Fig. 8 demonstrates that different latent spaces exhibit nearly identical mean norm upper bounds, thus justifying our focus on analyzing the number of modes $K$.

Given $\epsilon$ in Theorem A.5 and based on Assumption A.3, Eq. (10), Eq. (11), Eq. (13), we further have

$$
\begin{aligned}
\mathbb{E}[\|s_{\theta_t}(\mathbf{x}_t) - \nabla_{\mathbf{x}_t} \log p_t(\mathbf{x}_t)\|^2] &= \mathbb{E}\left[\left\|\sum_{i=1}^{K} \left(w_{i,t}(\mathbf{x}_t)\boldsymbol{\mu}_{i,t} - w_{i,t}^*(\mathbf{x}_t)\boldsymbol{\mu}_{i,t}^*\right)\right\|^2\right] \\
&\leq 2\mathbb{E}\left[\left\|\sum_{i=1}^{K} w_{i,t}^*(\mathbf{x}_t)(\boldsymbol{\mu}_{i,t} - \boldsymbol{\mu}_{i,t}^*)\right\|^2\right] + 2\mathbb{E}\left[\left\|\sum_{i=1}^{K} (w_{i,t}(\mathbf{x}_t) - w_{i,t}^*(\mathbf{x}_t))\boldsymbol{\mu}_{i,t}\right\|^2\right] \\
&\lesssim e^{-2t}(\epsilon^2 + B^2)
\end{aligned}
\tag{18}
$$

The $\lesssim$ hides constant term 2 and 4.

Therefore, consider a step size $h_k \leq \gamma$, we can have the learned score function $s_{\theta_t}(\mathbf{x})$ satisfies

$$\frac{1}{T} \sum_{k=1}^{N} h_k \mathbb{E}[\|s_{\theta_{t_k}}(\mathbf{x}_{t_k}) - \nabla_{\mathbf{x}_{t_k}} \log p_t(\mathbf{x}_{t_k})\|^2] \lesssim \frac{N\gamma}{T}(\epsilon^2 + B^2) \tag{19}$$

**Step 2: From Training Loss to Samlping Error.** In the practical sampling process, we adopt an early stopping strategy to improve the generation quality. Specifically, we consider the interval $t \in [0, 0.8]$ during the reverse process. Then, the following conclusion holds:

**Theorem A.6.** *(Theorem 2.2. in (Chen et al., 2023)) There is a universal constant $C$ such that the following hold. Suppose that Assumption A.3 and Eq. (19) hold and the step sizes satisfy the following for some quantities $\sigma_{t_1}^2, \ldots, \sigma_{t_k}^2, \ldots, \sigma_{t_N}^2$,*

$$\frac{h_k}{\sigma_{t_{k-1}}^2} \leq \frac{1}{Cd} \leq \gamma, \quad k = 1, \ldots, N \tag{20}$$

*Define $\Pi := \sum_{k=1}^{N} \frac{h_k^2}{\sigma_{t_{k-1}}^4}$. For $T \geq 2, \delta \leq \frac{1}{2}$, the exponential integrator scheme (6) with early stopping results in a distribution $\hat{q}_{T-\delta}$ such that*

$$\mathrm{KL}(p_\delta \| \hat{q}_{T-\delta}) \lesssim (d + B^2) \exp(-T) + T\epsilon_0^2 + d^2\Pi. \tag{21}$$

In particular, when using proper choices of $h_k$, the quantity $\Pi$ can be as small as $o(1)$. For instance, as shown in Chen et al. (2023), it can be proved that $\Pi = O(1/N^2)$ when using exponentially decreasing stepsize.

Then combing Theorem A.5 and Theorem A.6, we finally have

**Theorem A.7.** *Training DDPM for $H \geq H'$ iterations and uses $n \geq n'$ number of samples where*

$$H' = \Theta(\log(\varepsilon^{-1} \log d)), \quad n' = \Theta(K^4 d^5 B^6 / \varepsilon^2).$$

*Then, there is a universal constant $C$ such that the following hold. Suppose that Assumptions A.3 and Equation 19 hold and the step sizes satisfy*

$$\frac{h_k}{\sigma_{t_{k-1}}^2} \leq \frac{1}{Cd} \leq \gamma, \quad k = 1, \ldots, N \tag{22}$$

*Define $\Pi := \sum_{k=1}^{N} \frac{h_k^2}{\sigma_{t_{k-1}}^4}$. For $T \geq 2, \delta \leq \frac{1}{2}$, the exponential integrator scheme (6) with early stopping results in a distribution $\hat{q}_{T-\delta}$ such that*

$$\mathrm{KL}(p_\delta \| \hat{q}_{T-\delta}) \lesssim (d + B^2) \exp(-T) + N\gamma(\epsilon^2 + B^2) + d^2\Pi. \tag{23}$$

*where $p$ is the data distribution and $\hat{q}$ is the sampling distribution.*

In Theorem 2.1, we establish a connection between the training process and the sampling process, using KL-divergence as a metric to quantify the distance between the true data distribution and the sampled generated data distribution. It should be noted that both KL divergence and Wasserstein Distance serve as tools for measuring the similarity between distributions. Under the specific assumption that the data distributions are Gaussian, the Wasserstein Distance reduces to FID (i.e., the metric used in our paper). Theorem 2.1 demonstrates that achieving the same sampling error necessitates a larger number of training samples for data distributions with a greater number of modes ($K$). Consequently, under the constraint of limited training samples, the quality of images generated from training data distributions with more modes ($K$) tends to be worse compared to those with fewer modes.

## B. Experiments Setup

### B.1. Training Details of AEs

We present the training details of MAETok in Table 7.

### B.2. Training Details of Diffusion Models

We present the training details of SiT-XL and LightningDiT in Tables 8 and 9, which mainly follows their original setup.

### B.3. Training Details of GMM Models

In Fig. 2, we train our own AE, KL-VAE, and MAETok under exactly the same settings and use the pre-trained VAVAE (Yao & Wang, 2025). The evaluation in Fig. 2 is performed with the same latent size and input dimensions. Specially, for GMM in Fig. 2a, we first represent the original latent size as $(N, H, C)$, where $N$ refers to the training sample size, $H$ refers to the number of tokens, and $C$ refers to the channel size. Following the typical GMM training, we performed the following steps: (1) Latents flatten: The latent size becomes $(N, H \times C)$. (2) Dimensionality Reduction: To avoid the curse of dimensionality, we consider PCA and select a fixed dimension $K$ that results in an explained variance greater than 90%. This step makes the latent dimension $(N, K)$, ensuring that all latent spaces have consistent dimensions. (3) Normalization: To avoid numerical instability and feature scale differences, we further standardize the latent data. (4) Fitting: We fit the data using GMM and return the negative log-likelihood losses (NLL). We train the GMM on the entire Imagenet with a

| Configuration | Value |
|---|---|
| image resolution | 256×256, 512×512 |
| enc/dec hidden dimension | 768 |
| enc/dec #heads | 12 |
| enc/dec #layers | 12 |
| enc/dec patch size | 16 |
| enc/dec positional embedding | 2D RoPE (image), 1D APE (latent) |
| optimizer | AdamW (Loshchilov, 2017) |
| base learning rate | $1e^{-4}$ |
| weight decay | $1e^{-4}$ |
| optimizer momentum | $\beta_1, \beta_2 = 0.9, \ 0.95$ |
| global batch size | 512 |
| learning rate schedule | cosine |
| warmup steps | 10K |
| training steps | 500K |
| augmentation | horizontal flip, center crop |
| discriminator | DINOv2-S |
| discriminator weight | 0.4 with adaptive weight |
| discriminator start | 30K |
| discriminator LeCAM | 0.001 (Tseng et al., 2021) |
| perceptual weight | 1.0 |
| evaluation metric | FID-50k |

*Table 7.* Training configuration of MAETok on 256×256 and 512×512 ImageNet.

batch size of 256 on a single NVIDIA A8000. It should be noted that distributed training would further optimize the fitting time. The training time for GMM components of 50, 100, and 200 is roughly 3, 8, and 11 hours, respectively.

For SiT-L loss in Fig. 2b, we train SiT-L on the latent space of these four tokenizers for 400K iterations, using an optimizer of AdamW, a constant learning rate of 1e-4, and no weight decay.

## C. Experiments Results

### C.1. More Quantitative Generation Results

We provide the additional precision and recall evaluation on 256×256 and 512×512 ImageNet benchmarks in Table 10 and Table 11, respectively.

### C.2. Classifier-free Guidance Tuning Results

We provide our CFG scale tuning results in Table 12, where we found the gFID with CFG changes significantly even with small guidance scales. Applying CFG interval (Kynkäänniemi et al., 2024) to cutout the high timesteps with CFG can mitigate this issue. However, it is still extremely difficult to tune the guidance scale.

We use a guidance scale of 1.9 and an interval of [0, 0.75] for 256×256 SiT-XL and a guidance scale of 1.8 and an interval of [0, 0.75] for 256×256 LightningDiT to report the main results. For 512× models, we use a guidance scale of 1.5 and an interval of [0, 0.7] for SiT-XL and a guidance scale of 1.6 with an interval of [0, 0.65] for LightningDiT's main results. *Note that our models may present even better results with more fine-grained CFG tuning.*

We attribute the difficulty of tuning CFG to the semantics learned by the unconditional class, as we discussed in Section 4.5. Such semantics makes the linear scheme of CFG less effective, as reflected by the sudden change with small guidance values. Adopting and designing more advanced CFG schemes (Chung et al., 2024; Karras et al., 2024) may also be helpful with this problem, and is left as our future work.

### C.3. Latent Space Visualization

More latent space visualization of MAETok variants is included in Fig. 9. MAETok in general learns more discriminative latent space with fewer GMM models with differente reconstruction targets.

| Configuration | Value |
|---|---|
| image resolution | 256×256, 512×512 |
| hidden dimension | 1152 |
| #heads | 16 |
| #layers | 28 |
| patch size | 1 |
| positional embedding | 1D sinusoidal |
| optimizer | AdamW (Loshchilov, 2017) |
| base learning rate | $1e^{-4}$ |
| weight decay | 0.0 |
| optimizer momentum | $\beta_1, \beta_2 = 0.9,\ 0.999$ |
| global batch size | 256 |
| learning rate schedule | constant |
| training steps | 4M |
| augmentation | horizontal flip, center crop |
| diffusion sampler | Euler-Maruyama |
| diffusion steps | 250 |
| evaluation suite | ADM (Dhariwal & Nichol, 2021) |
| evaluation metric | FID-50k |

*Table 8.* Training configuration of SiT-XL on 256×256 and 512×512 ImageNet.

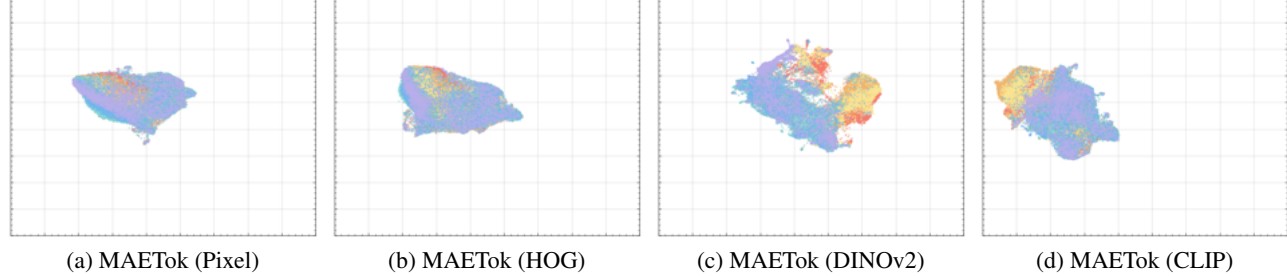

| (a) MAETok (Pixel) | (b) MAETok (HOG) | (c) MAETok (DINOv2) | (d) MAETok (CLIP) |

*Figure 9.* UMAP visualization on ImageNet of the learned latent space from (a) MAETok with raw pixel target; (b) MAETok with HOG target; (c) MAETok with DINOv2 target; (d) MAETok with CLIP target. MAETok presents a more discriminative latent space.

## C.4. More Ablation Results

We present the ablation study on latent tokens and 2D RoPE in Table 13. One can observe from Table 13a that using learnable latent tokens is more effective than using image tokens only, and 128 latent tokens is enough to achieve similar reconstruction and downstream generation performance, compared to 256 tokens. Furthermore, 2D RoPE helps to generalize better on different resolutions, when trained with mixed resolution images.

## C.5. More Qualitative Generation Results

| Configuration | Value |
|---|---|
| image resolution | 256×256, 512×512 |
| hidden dimension | 1152 |
| #heads | 16 |
| #layers | 28 |
| patch size | 1 |
| positional embedding | 1D RoPE |
| optimizer | AdamW (Loshchilov, 2017) |
| base learning rate | $2e^{-4}$ |
| weight decay | 0.0 |
| optimizer momentum | $\beta_1, \beta_2 = 0.9, \ 0.95$ |
| global batch size | 1024 |
| learning rate schedule | constant |
| training steps | 400K |
| augmentation | horizontal flip, center crop |
| additional loss | cosine loss |
| diffusion sampler | Euler |
| diffusion steps | 250 |
| evaluation suite | ADM (Dhariwal & Nichol, 2021) |
| evaluation metric | FID-50k |

*Table 9.* Training configuration of LightningDiT on 256×256 and 512×512 ImageNet.

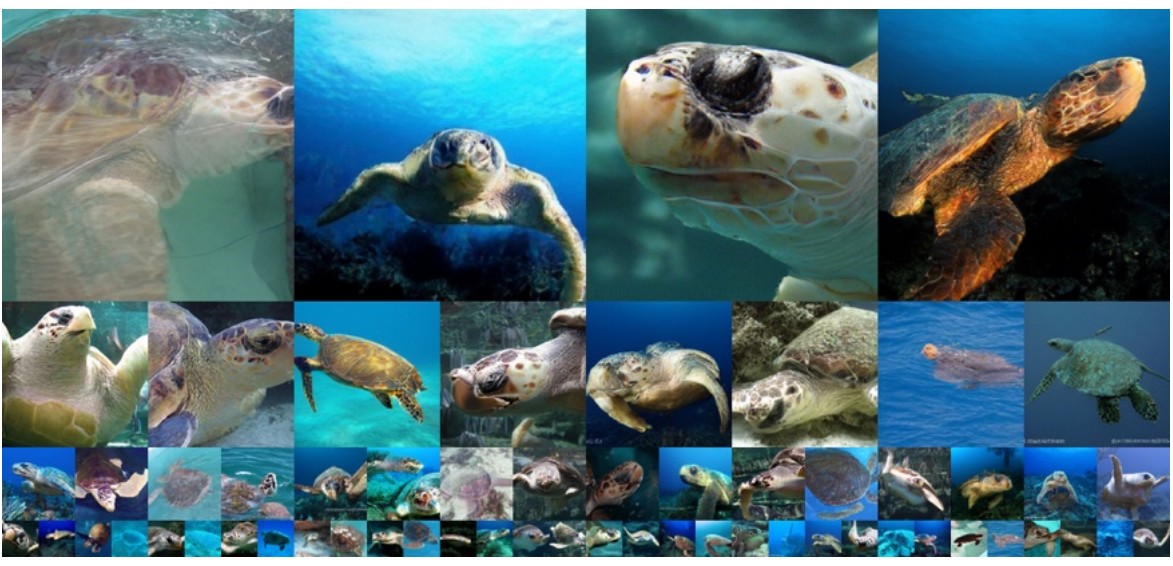

*Figure 10.* Uncurated generation results of 256×256 MAETok + SiT-XL. We use CFG of 3.0. Class label = "Loggerhead" (33).

| Model (G) | # Params (G) | Model (T) | # Params (T) | # Tokens↓ | rFID↓ | w/o CFG | | | | w/ CFG | | | |
|---|---|---|---|---|---|---|---|---|---|---|---|---|---|
| | | | | | | gFID↓ | IS↑ | Prec↑ | Recall↑ | gFID↓ | IS↑ | Prec↑ | Recall↑ |
| *Auto-regressive* | | | | | | | | | | | | | |
| VQGAN (Esser et al., 2021) | 1.4B | VQ | 23M | 256 | 7.94 | – | – | – | – | 5.20 | 290.3 | – | – |
| ViT-VQGAN (Yu et al., 2021) | 1.7B | VQ | 64M | 1024 | 1.28 | 4.17 | 175.1 | – | – | – | – | – | – |
| RQ-Trans. (Lee et al., 2022) | 3.8B | RQ | 66M | 256 | 3.20 | – | – | – | – | 3.80 | 323.7 | – | – |
| MaskGIT (Chang et al., 2022) | 227M | VQ | 66M | 256 | 2.28 | 6.18 | 182.1 | 0.80 | 0.51 | – | – | – | – |
| MAGE (Li et al., 2023) | 439M | VQ | (N/A) | 256 | – | 6.93 | 195.8 | – | – | – | – | – | – |
| LlamaGen-3B (Sun et al., 2024) | 3.1B | VQ | 72M | 576 | 2.19 | – | – | – | – | 2.18 | 263.3 | 0.80 | 0.58 |
| TiTok-S-128 (Yu et al., 2024c) | 287M | VQ | 72M | 128 | 1.61 | – | – | – | – | 1.97 | 281.8 | – | – |
| VAR (Tian et al., 2024) | 2B | MSRQ† | 109M | 680 | 0.90 | – | – | – | – | 1.92 | 323.1 | 0.82 | 0.60 |
| ImageFolder (Li et al., 2024c) | 362M | MSRQ | 176M | 286 | 0.80 | – | – | – | – | 1.92 | 323.1 | 0.75 | 0.63 |
| MAGVIT-v2 (Yu et al., 2024a) | 307M | LFQ | 116M | 256 | 1.61 | 3.07 | 213.1 | – | – | 1.78 | 319.4 | – | – |
| MaskBit (Weber et al., 2024) | 305M | LFQ | 54M | 256 | 1.61 | – | – | – | – | 1.52 | 328.6 | – | – |
| MAR-H (Li et al., 2024b) | 943M | KL | 66M | 256 | 1.22 | 2.35 | 227.8 | 0.79 | 0.62 | 1.55 | 303.7 | 0.81 | 0.62 |
| *Diffusion-based* | | | | | | | | | | | | | |
| LDM-4 (Rombach et al., 2022b) | 400M | KL† | 55M | 4096 | 0.27 | 10.56 | 103.5 | 0.71 | 0.62 | 3.60 | 247.7 | 0.87 | 0.48 |
| U-ViT-H/2 (Bao et al., 2023) | 501M | | | | | – | – | – | – | 2.29 | 263.9 | 0.82 | 0.57 |
| MDTv2-XL/2 (Gao et al., 2023) | 676M | | | | | 5.06 | 155.6 | 0.72 | 0.66 | 1.58 | 314.7 | 0.79 | 0.65 |
| DiT-XL/2 (Peebles & Xie, 2023) | 675M | KL† | 84M | 1024 | 0.62 | 9.62 | 121.5 | 0.67 | 0.67 | 2.27 | 278.2 | 0.83 | 0.53 |
| SiT-XL/2 (Ma et al., 2024) | 675M | | | | | 8.30 | 131.7 | 0.68 | 0.67 | 2.06 | 270.3 | 0.82 | 0.59 |
| + REPA (Yu et al., 2024d) | 675M | | | | | 5.90 | 157.8 | 0.70 | 0.69 | 1.42 | 305.7 | 0.80 | 0.65 |
| TexTok-256 (Zha et al., 2024) | 675M | KL | 176M | 256 | 0.69 | – | – | – | – | 1.46 | 303.1 | 0.79 | 0.64 |
| LightningDiT (Yao & Wang, 2025) | 675M | KL† | 70M | 256 | 0.28 | 2.17 | 205.6 | – | – | 1.35 | 295.3 | – | – |
| *Ours* | | | | | | | | | | | | | |
| MAETok + LightningDiT | 675M | AE | 176M | **128** | 0.48 | 2.21 | 208.3 | 0.79 | 0.62 | 1.73 | 308.4 | 0.80 | 0.63 |
| MAETok + SiT-XL | 675M | | | | | 2.31 | 216.5 | 0.78 | 0.62 | 1.62 | 310.6 | 0.81 | 0.63 |

*Table 10.* System-level comparison on ImageNet 256×256 conditional generation, now also reporting Precision and Recall under both CFG and no-CFG settings. "Model (G)": generation model. "# Params (G)": the number of generator parameters. "Model (T)": the tokenizer model. "# Params (T)": the number of tokenizer parameters. "# Tokens": the number of latent tokens used during generation. † indicates that the model has also been trained on data beyond ImageNet.

| Model (G) | # Params (G) | Model (T) | # Params (T) | # Tokens↓ | rFID↓ | w/o CFG | | | | w/ CFG | | | |
|---|---|---|---|---|---|---|---|---|---|---|---|---|---|
| | | | | | | gFID↓ | IS↑ | Prec↑ | Recall↑ | gFID↓ | IS↑ | Prec↑ | Recall↑ |
| *GAN* | | | | | | | | | | | | | |
| BigGAN (Chang et al., 2022) | – | – | – | – | – | – | – | – | – | 8.43 | 177.9 | – | – |
| StyleGAN-XL (Karras et al., 2019) | 168M | – | – | – | – | – | – | – | – | 2.41 | 267.7 | – | – |
| *Auto-regressive* | | | | | | | | | | | | | |
| MaskGIT (Chang et al., 2022) | 227M | VQ | 66M | 1024 | 1.97 | 7.32 | 156.0 | – | – | – | – | – | – |
| TiTok-B-64 (Yu et al., 2024c) | 177M | VQ | 202M | 128 | 1.52 | – | – | – | – | 2.13 | 261.2 | – | – |
| MAGVIT-v2 (Yu et al., 2024a) | 307M | LFQ | 116M | 1024 | - | – | – | – | – | 1.91 | 324.3 | – | – |
| MAR-H (Li et al., 2024b) | 943M | KL | 66M | 1024 | – | 2.74 | 205.2 | 0.69 | 0.59 | 1.73 | 279.9 | 0.77 | 0.61 |
| *Diffusion-based* | | | | | | | | | | | | | |
| ADM (Dhariwal & Nichol, 2021) | – | – | – | – | – | 23.24 | 58.06 | – | – | 3.85 | 221.7 | 0.84 | 0.53 |
| U-ViT-H/4 (Bao et al., 2023) | 501M | | | | | – | – | – | – | 4.05 | 263.8 | 0.84 | 0.48 |
| DiT-XL/2 (Peebles & Xie, 2023) | 675M | KL† | 84M | 4096 | 0.62 | 9.62 | 121.5 | – | – | 3.04 | 240.8 | 0.84 | 0.54 |
| SiT-XL/2 (Ma et al., 2024) | 675M | | | | | – | – | – | – | 2.62 | 252.2 | 0.84 | 0.57 |
| DiT-XL (Chen et al., 2024b) | 675M | | | | | 9.56 | – | – | – | 2.84 | – | – | – |
| UViT-H (Chen et al., 2024b) | 501M | | | | | 9.83 | – | – | – | 2.53 | – | – | – |
| UViT-H (Chen et al., 2024b) | 501M | AE† | 323M | 256 | 0.22 | 12.26 | – | – | – | 2.66 | – | – | – |
| UViT-2B (Chen et al., 2024b) | 2B | | | | | 6.50 | – | – | – | 2.25 | – | – | – |
| USiT-2B (Chen et al., 2024b) | 2B | | | | | 2.90 | – | – | – | 1.72 | – | – | – |
| *Ours* | | | | | | | | | | | | | |
| MAETok + LightningDiT | 675M | AE | 176M | **128** | 0.62 | **2.56** | **224.5** | – | – | 1.72 | 307.3 | 0.81 | 0.62 |
| MAETok + SiT-XL | 675M | | | | | 2.79 | 204.3 | 0.81 | 0.62 | **1.69** | 304.2 | 0.82 | 0.62 |

*Table 11.* System-level comparison on ImageNet 512×512 conditional generation, now also reporting Precision and Recall for both CFG and no-CFG settings. "Model (G)": generation model. "# Params (G)": number of generator parameters. "Model (T)": the tokenizer model. "# Params (T)": number of tokenizer parameters. "# Tokens": number of latent tokens used during generation. † indicates the model was also trained on data beyond ImageNet.

| CFG | 1.7 | 1.8 | 1.9 | 2.0 | 1.8 | 1.9 | 2.0 | 1.9 | 2.0 | 1.7 | 1.8 | 1.8 |
|---|---|---|---|---|---|---|---|---|---|---|---|---|
| Interval | [0, 0.7] | [0, 0.7] | [0, 0.7] | [0, 0.7] | [0, 0.75] | [0, 0.75] | [0, 0.75] | [0, 0.8] | [0, 0.8] | [0, 1.0] | [0, 1.0] | [0.125, 0.8] |
| gFID | 4.96 | 4.94 | 4.91 | 4.92 | 4.92 | 4.92 | 4.94 | 5.09 | 5.14 | 5.21 | 8.55 | 6.08 |
| IS | 267.87 | 275.87 | 282.52 | 288.78 | 290.47 | 299.36 | 306.31 | 318.41 | 326.97 | 304.58 | 349.97 | 289.27 |

*Table 12.* CFG tuning results of 256×256 SiT-XL trained for 2M steps. We compute gFID and IS using 10K generated samples.

| Tok | 1D | # Tokens | rFID | gFID |
|---|---|---|---|---|
| VAVAE | ✓ | 256 | 0.28 | 13.65 |
| MAETok | ✓ | 256 | 0.37 | 5.05 |
| MAETok | | 256 | 1.01 | 6.85 |
| MAETok | ✓ | 128 | 0.48 | 5.69 |

(a) Latent tokens.

| Pos. Emb. | 256 rFID | 512 rFID |
|---|---|---|
| APE | 0.73 | 1.43 |
| RoPE | 0.51 | 0.72 |

(b) RoPE.

*Table 13.* Ablations of latent tokens and 2D RoPE with MAETok on 256×256 ImageNet. We report rFID of tokenizer and gFID of SiT-L trained on latent space of the tokenizer without classifier-free guidance. We train tokenizer of 250K and SiT-L for 400K steps.

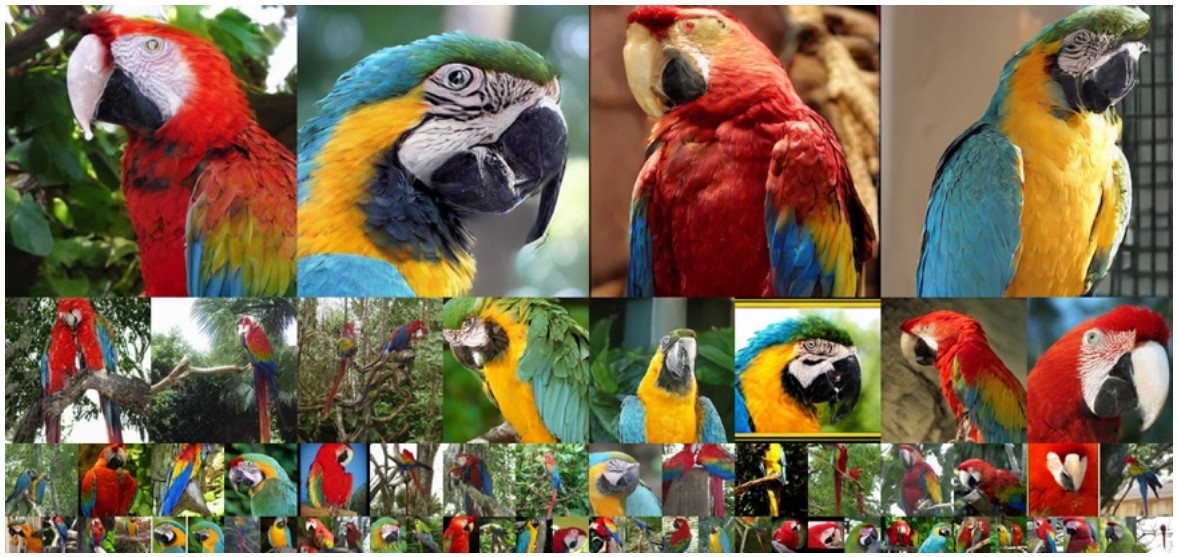

*Figure 11.* Uncurated generation results of 256×256 MAETok + SiT-XL. We use CFG of 3.0. Class label = "Macaw" (88).

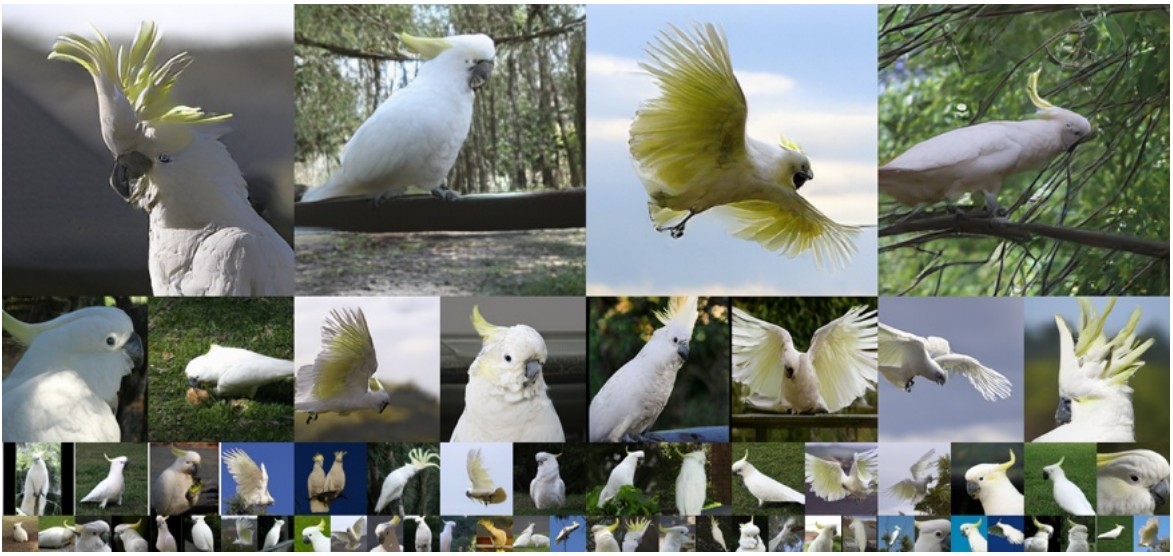

*Figure 12.* Uncurated generation results of 256×256 MAETok + SiT-XL. We use CFG of 3.0. Class label = "Cacatua galerita" (89).

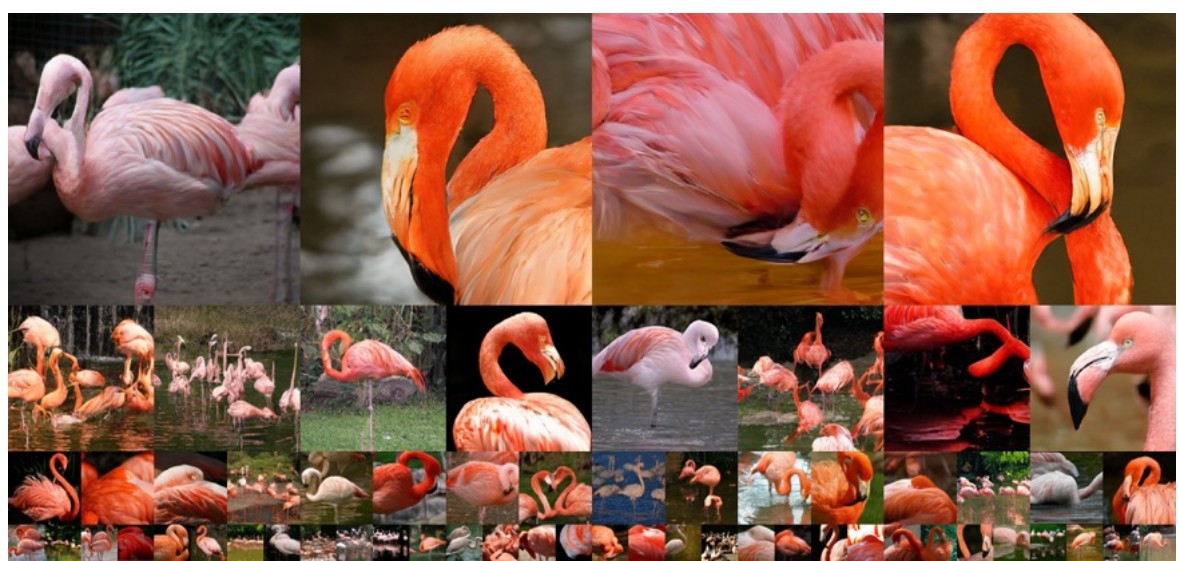

*Figure 13.* Uncurated generation results of 256×256 MAETok + SiT-XL. We use CFG of 3.0. Class label = "Flamingo" (130).

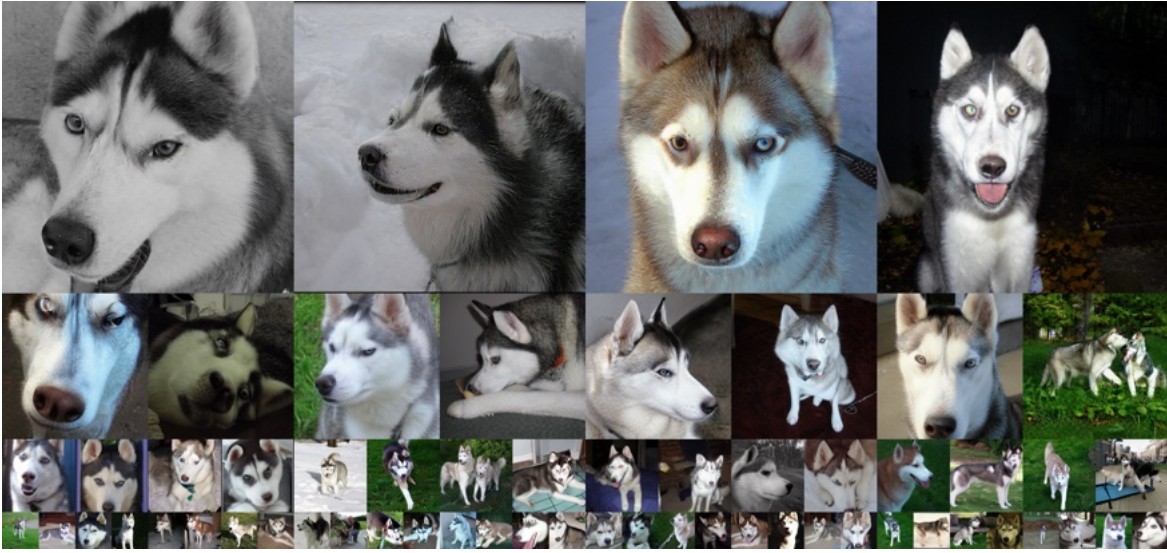

*Figure 14.* Uncurated generation results of 512×512 MAETok + SiT-XL. We use CFG of 2.0. Class label = "Siberian husky" (250).

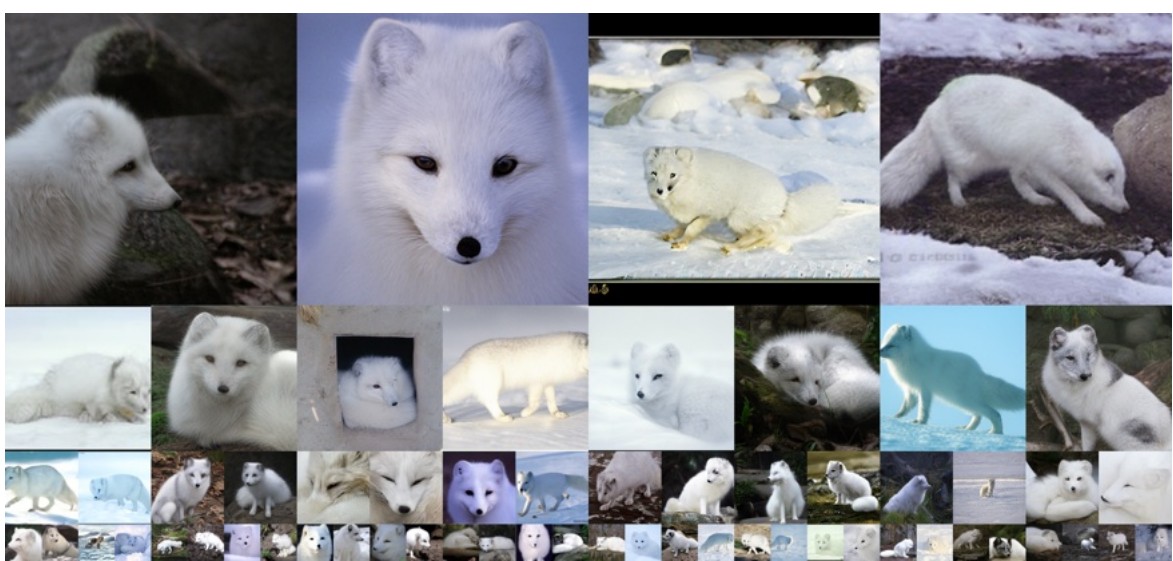

*Figure 15.* Uncurated generation results of 512×512 MAETok + SiT-XL. We use CFG of 2.0. Class label = "Arctic fox" (279).

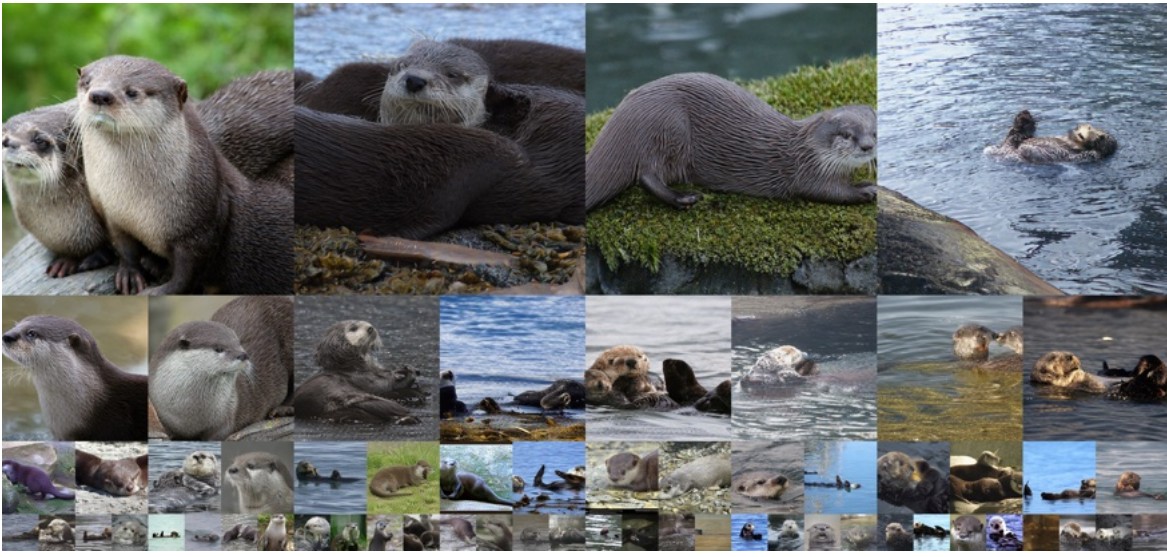

*Figure 16.* Uncurated generation results of 512×512 MAETok + SiT-XL. We use CFG of 2.0. Class label = "Otter" (360).

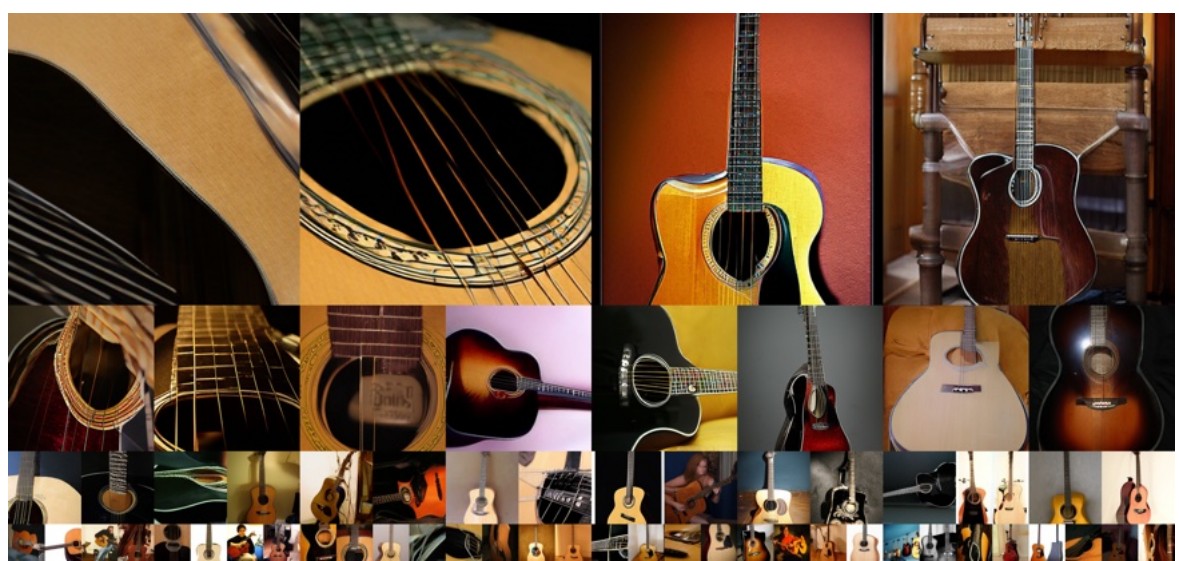

*Figure 17.* Uncurated generation results of 512×512 MAETok + SiT-XL. We use CFG of 2.0. Class label = "Guitar" (402).

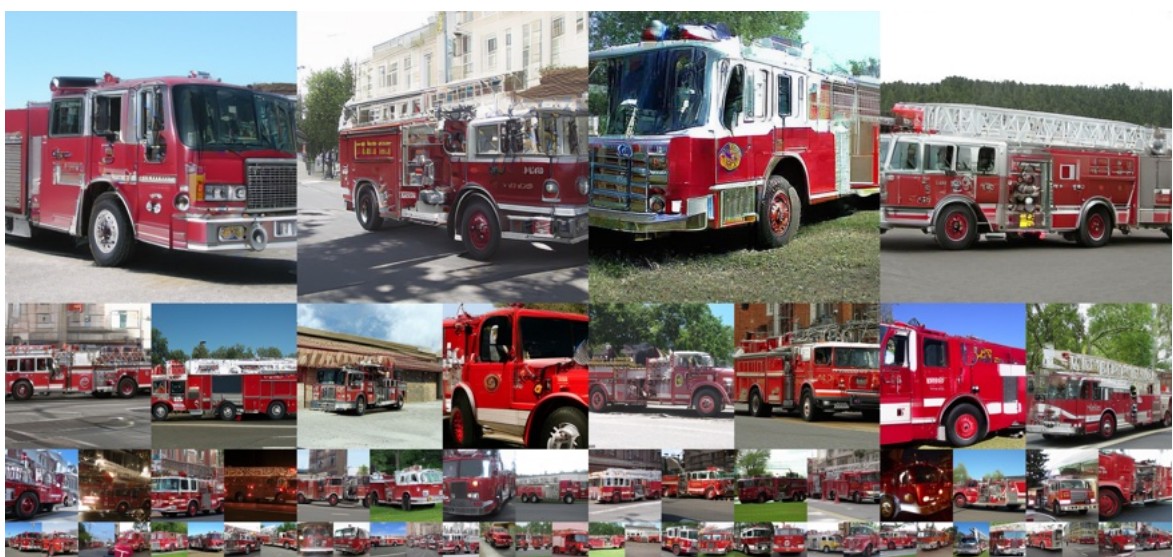

*Figure 18.* Uncurated generation results of 512×512 MAETok + SiT-XL. We use CFG of 2.0. Class label = "Fire Truck" (555).

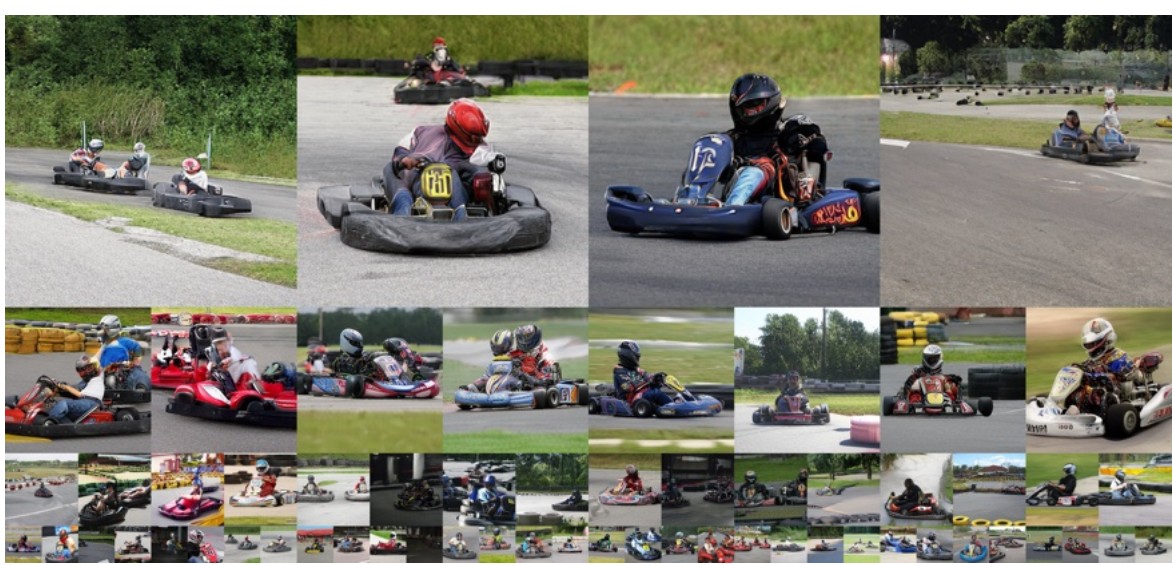

*Figure 19.* Uncurated generation results of 512×512 MAETok + SiT-XL. We use CFG of 2.0. Class label = "Go-kart" (573).

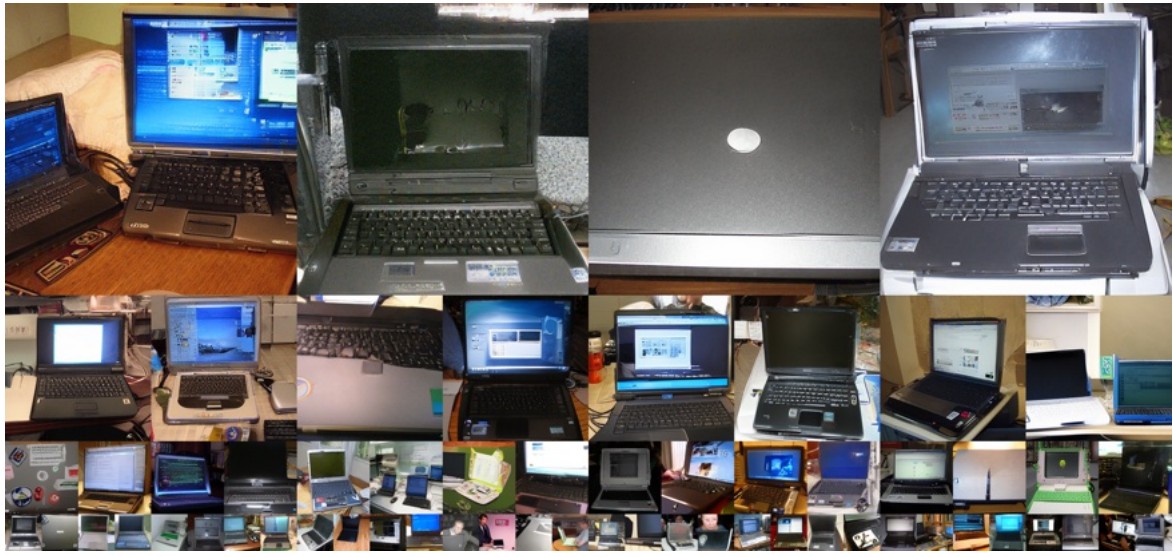

*Figure 20.* Uncurated generation results of 512×512 MAETok + SiT-XL. We use CFG of 2.0. Class label = "Laptop" (620).

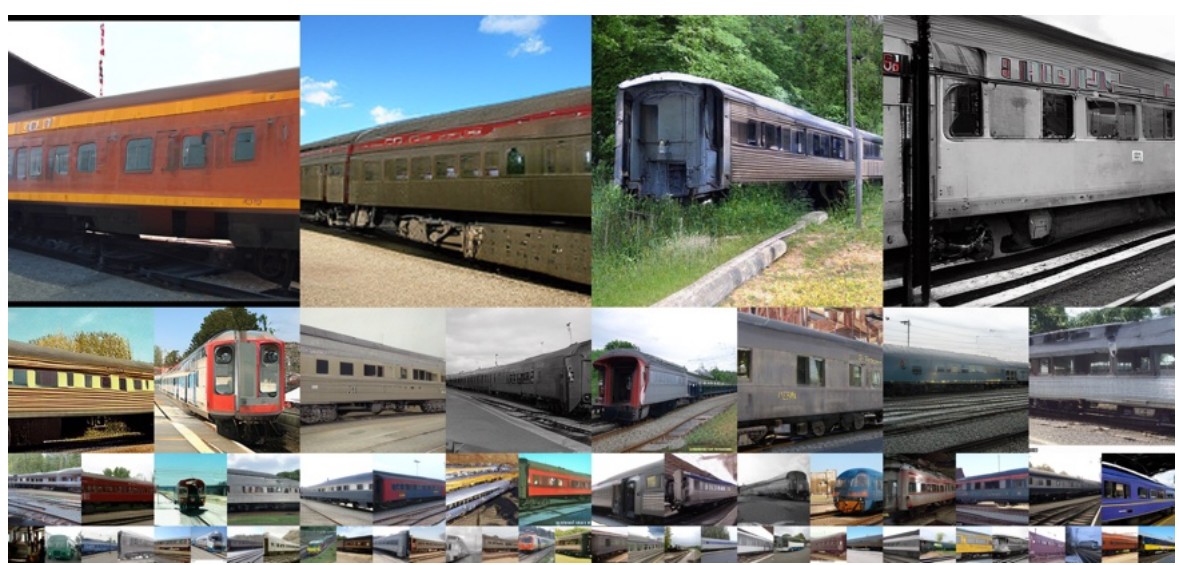

*Figure 21.* Uncurated generation results of 512×512 MAETok + SiT-XL. We use CFG of 2.0. Class label = "Carriage" (705).

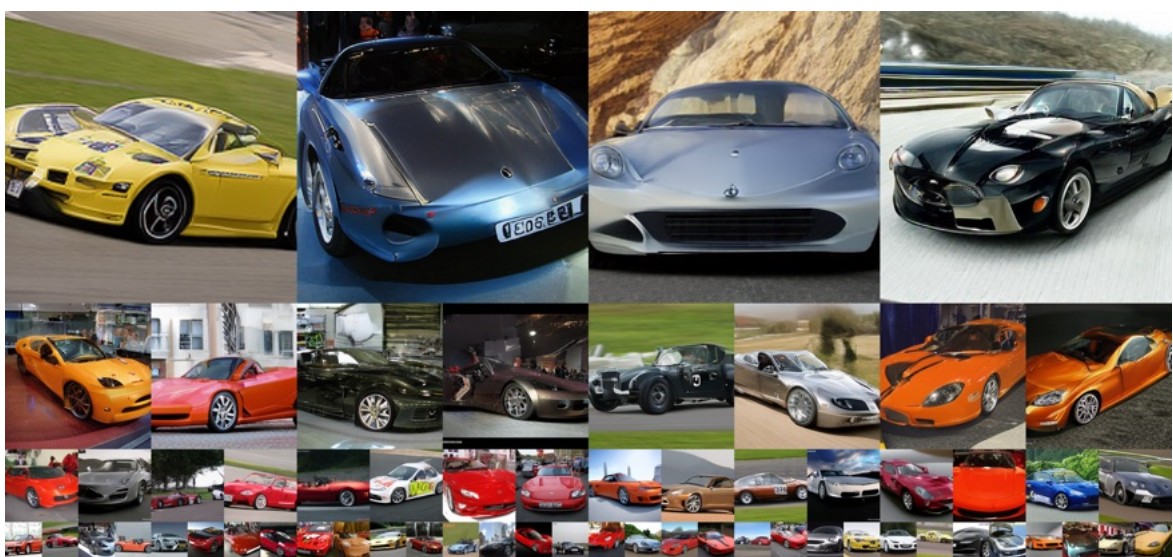

*Figure 22.* Uncurated generation results of 512×512 MAETok + SiT-XL. We use CFG of 2.0. Class label = "Sports Car" (402).

