# OpenReview forum: "Masked Autoencoders Are Effective Tokenizers for Diffusion Models"
_ICML.cc/2025/Conference — ICML 2025 spotlightposter_

### Official Review · Reviewer_cxaR · 2025-03-10

**Overall Recommendation:** 3

**Summary:**

This paper proposes to use masked autoencoder for reconstruction and verify it works better for diffusion model generation compared to AE and VAE.

**Claims And Evidence:**

It's a bit contradictive between figure 2 and 4, from Figure 4, AE seems to have fewer GMM mode, i.e. it's more concentrated on one mode.

**Essential References Not Discussed:**

.

**Experimental Designs Or Analyses:**

I expect the authors to elaborate details on how they get figure 2, what dataset do they use to get statistics of the latent space? Are the latent dimension the same for all the four methods?

What's the difference of ablation study in Figure 2 and Table 1a, Why not add VAVAE in table 1 comparison too?

Are you able to increase your model size and compare with DC-AE?

Can you some show ablation analysis on the 2D ROPE, is it useful?

**Methods And Evaluation Criteria:**

What’s the motivation to introduce learnable token z? What would happen if you do not introduce learnable token z?

I think generally this paper proposes several techniques: 1. masked autoencoder, 2. learnable token; 3. auxiliary decoders to align latent feature. I'd like to know what's their weight in the contribution. For example, if you remove 3 and keep 1 and 2, how much quality would drop, and if you remove 1 and keep 2 and 3 what would happen, etc.

**Other Comments Or Suggestions:**

It’s better if you can explain VAVAE a bit before diving into that around Figure 2

**Other Strengths And Weaknesses:**

.

**Questions For Authors:**

.

**Relation To Broader Scientific Literature:**

.

**Theoretical Claims:**

No

---

> ### Author Rebuttal · Authors · 2025-04-01
>
> We thanks for your time reviewing this paper and suggestions on additional ablation studies and comparison results.
>
>
> ---
>
> > It's a bit contradictive between figure 2 and 4.
>
> Thanks for your question.
>
> - **Figure 4 and Figure 2 are aligned**. Figure 4 shows that the latent space of AE is more concentrated compared to others. In a concentrated latent space, increasing the number of modes in the GMM may not significantly improve fitting performance (e.g., reduce NLL) due to the close distances between modes, as observed in Figure 2.
> -  **Compactness in two dimensions (UMAP) is not highly correlated to its discriminative and modes**. To support our claim, we conduct a [synthetic experiment](https://anonymous.4open.science/api/repo/Rebuttal-8656/file/ID14576_%20Synthetic.pdf?v=def2b7f2) with two Gaussian Mixtures: A (30 close modes) and B (4 distant modes). UMAP shows that data with fewer modes (B) is more spread out, while data with more modes (A) is more compact.
>
> > Motivation of learnable token z?
>
> The motivation of learnable token is for (1) more flexibility compression of images, and thus for more efficient downstream generative model training as discussed Sec. 4.5, and (2) for more compression purpose of MAE learning.
> The following is an additional comparison on learnable tokens of the tokenizer and downstream SiT-L performance.
>
> | Latent  | # Tokens  | rFID  | gFID  |
> |:---:|:---:|:---:|:---:|
> | Learnable Token  | 128  |  0.85 | 5.78  |
> | Image Token  | 256  | 1.01  | 6.85  |
>
> We will add this comparison in our revised paper.
>
> >  Ablation on 1. masked autoencoder, 2. learnable token; 3. auxiliary decoders.
>
> We have provided ablation study mainly focusing on 1 (masked autoencoder) and 3 (auxiliary decoders) in Table 1.
> Here, we additional provide some ablation results on 2 (learnable tokens):
>
> | 1. masked autoencoder  | 2. learnable tokens  | 3. auxiliary decoders  | rFID  | gFID  |
> |:---:|:---:|:---:|:---:|:---:|
> | 1  | 1  | 1  | 0.85  | 5.78  |
> | 0  | 1  | 1  | 0.64  | 8.44  |
> | 1  | 0  | 1  | 1.01  | 6.85  |
> | 1  | 1  | 0  | 1.15  | 17.18  |
> | 0  | 0  | 1  | 0.43  | 9.88 |
> | 1  | 0  | 0  | 0.96    | 18.23 |   |
> | 0  | 1  | 0  | 0.67  | 24.47  |
>
> From the results, removing learnable tokens may lead to both decrease of rFID and gFID.
> Masking modeling is necessary for learning better latent space other than using only auxiliary decoders.
> We will include these ablation results in our revised paper.
>
> > Elaborate details on figure 2.
>
> Sorry for the confusion. We train AE, KL-VAE, and MAETok under the same settings and use the pre-trained VAVAE. The analysis is performed with the same latent size. Specifically:
> - **Latents Flatten** and **Dimensionality Reduction**: The flatten operation first changes the latent size from $(N, H, C)$ to $(N, H \times C)$, then the dimension is reduced to $(N, K)$, where $K$ is the dimension that explains over 90% of the variance and ensures same latent sizes.
> - **Normalization** and **Fitting**: We standardize the latent data and then fit the GMM model.
>
> > Difference of ablation study in Figure 2 and Table 1a, Why not add VAVAE in table 1 comparison too?
>
> Figure 2 and Table 1 are under exact the same settings.
> We did not include VAVAE mainly because space consideration, i.e. 5 rows for each subtable.
> MAETok do outperform VAVAE, as shown by gFID results in Figure 2 and the SiT-L generation results below:
>
> | Tokenizer  | # Tokens  | LP  | rFID  | gFID  |
> |:---: |:---: |:---: |:---: |:---: |
> | MAETok  | 128  |  72.3 | 0.48  | 5.69  |
> | VAVAE  | 256  |  54.1 |  0.28 |  13.65 |
>
> We will add this to Table 1 for less confusion.
>
> > Are you able to increase your model size and compare with DC-AE?
>
> We provide the 512x512 generation results of training 2B USiT with MAETok for 500K steps (as in DC-AE) below:
>
> | Tokenizer  | # Params | # Tokens  | rFID  | gFID w/o CFG  | gFID w/ CFG  |
> |:---:|:---:|:---:|:---:|:---:|:---:|
> | MAETok | 176M | 128  | 0.48  | 1.72 | 1.65 |
> | DC-AE  | 323M | 256  |  0.2s2 |  2.90 | 1.72 |
>
> **MAETok with only 128 tokens established a new SOTA on 512x512 generation**: our gFID w/o CFG already outperforms previous results with CFG.
> We will include this comparison in our revised paper.
>
> > Ablation analysis on the 2D ROPE?
>
> 2D ROPE allows easier and better tokenization with mixed resolution images.
> We fine-tune MAETok with ROPE and absolute position embedding (APE) on mixed 256x256 and 512x512 images, where 2D ROPE not only presents better results but also generalizes better on higher resolution.
>
> | Position Embedding  | 256x256 rFID  | 512x512 rFID  |
> |:---: |:---: |--- |
> | 2D ROPE  | 0.51  | 0.72  |
> | 2D APE  | 0.73  | 1.43  |
>
>
> > It’s better if you can explain VAVAE a bit before diving into that around Figure 2
>
> We will add these results in our revised paper.
>
> ---
>
> We hope the above results could resolve your concern and further validate the effectiveness of MAETok.
> If you also find the results helpful, please consider raise our rating, thanks!

---

### Official Review · Reviewer_t7ec · 2025-03-12

**Overall Recommendation:** 3

**Summary:**

This paper analyzes how to develop a good image tokenizer. The authors bridge the GMM model and the quality of the latent space for generation and provide interesting discussion. Based on their investigation, they introduce MAETok to regularize the latent space with mask modeling when training tokenizer and achieve promising generation results.

**Claims And Evidence:**

This paper should provide more detailed experiments to support its claims.

1) Does the evaluation in Figure 2 perform at the same latent size? The authors should provide more detailed experimental settings about this experiment.
2) Can the finding in Figure 2 be a direct criterion to find a good tokenizer? How much time/computation resource is needed to fit the GMM model, compared to directly training a downstream generative model?
3) Could you provide a comparison of semantic regularization added in training the tokenizer versus training the generative model (e.g., REPA)?
4) Could you also provide an aligned latent size (e.g., 128 tokens, 256 tokens) for better comparison with previous works, since your contribution of MAE is not directly related to latent size?

**Essential References Not Discussed:**

The authors should discuss with 'Rethinking the Objectives of Vector-Quantized Tokenizers for Image Synthesis,' which examines the dillema between reconstruction and generation in vector-quantized tokenizers.

**Experimental Designs Or Analyses:**

Yes.

**Methods And Evaluation Criteria:**

Yes.

**Other Comments Or Suggestions:**

N/A.

**Other Strengths And Weaknesses:**

This paper reveals an interesting finding about the relationship between GMM mode and downstream generation performance. However, it has weaknesses in experimental design. See the "Claims and Evidence" part for my detailed comments on adding experiments to support the claims.

**Questions For Authors:**

N/A.

**Relation To Broader Scientific Literature:**

This paper relates to a broader effort to design tokenizers and find good latent spaces for generative models.

**Theoretical Claims:**

I have checked the correctness of theoretical claims.

---

> ### Author Rebuttal · Authors · 2025-04-01
>
> Thanks for your time and efforts reviewing our paper. We now address the raised questions as follows.
>
>
> ---
>
> > Does the evaluation in Figure 2 perform at the same latent size? The authors should provide more detailed experimental settings about this experiment.
>
> Sorry for the confusion. In Figure 2, we train our own AE, KL-VAE, and MAETok under exactly the same settings and use the pre-trained VAVAE. The evaluation in Figure 2 is performed with the same latent size and input dimensions. Specially,
>
> (1) For GMM in **Figure 2(a)**
>
> We first represent the original latent size as $(N, H, C)$, where $N$ refers to the training sample size, $H$ refers to the number of tokens, and $C$ refers to the channel size. Following the typical GMM training, we performed the following steps:
> - **Latents flatten**: The latent size becomes $(N, H \times C)$.
> - **Dimensionality Reduction**: To avoid the curse of dimensionality, we consider PCA and select a fixed dimension $K$ that results in an explained variance greater than 90%. This step makes the latent dimension $(N,K)$, ensuring that all latent spaces have consistent dimensions.
> - **Normalization**: To avoid numerical instability and feature scale differences, we further standardize the latent data.
> - **Fitting**: We fit the data using GMM and return the negative log-likelihood losses (NLL).
>
>
> (2) For SiT-L loss in **Figure 2(b)**
> - We train SiT-L on the latent space of these four tokenizers for 400K iterations, using an optimizer of AdamW, a constant learning rate of 1e-4, and no weight decay.
>
>
> We will include these experiment details in our revised Appendix.
>
>
> > Can the finding in Figure 2 be a direct criterion to find a good tokenizer? How much time/computation resource?
>
>
> Thanks for this good question. In our image generation setting, if the decoders of different tokenizers present similar capability (rFID), the finding in Figure 2 is can be used to verify on good encoder/latent space for downstream generative models. For broader tokenizers, such as those for multimodal or video data, whether the findings in Figure 2 are applicable requires further discussion. We will leave additional exploration of this for future work.
>
>
> The time and computation for the GMM analysis of the latent space is cheap. Using the Tokenizer AE, we train the GMM on the entire Imagenet with a batch size of 256 on a single NVIDIA A8000 GPU. It should be noted that distributed training would further optimize the fitting time. We consider various components and report corresponding times as follows:
>
>
> | #Components| Time (h) |
> | -------- | -------- |
> | 50  |   3  |
> | 100  |  8   |
> | 200  |   11   |
>
> The GMM analysis time is much less compared to the downstream generative model training time.
> For example, training SiT-XL for 4M steps on 8xH100 GPUs takes at least one week.
>
>
> > Could you provide a comparison of semantic regularization tokenizer versus generative model (e.g., REPA)?
>
>
> Thanks for this interesting question.
> - First, a gFID convergence comparison of semantic regularization tokenizers, e.g. MAETok, with REPA is already included in Figure 5b. We also have the system-level comparison of training plain SiT-XL on MAETok, with REPA, MDTv2 in Table 2 and Table 4, which are semantic regularization methods added into training of generative models. These results show that **adding semantic information to the tokenizer outperforms adding semantic information to the generative model, as evidenced by the smaller gFID**. This comparison demonstrates a more fundamental improvement from the latent space than generative models.
> - Secondly, **semantic regularization in the generative model and in the tokenizer are not mutually exclusive.** We added additional experiments with REPA and MAETok and observed that the gFID of training SiT-L on MAETok decreased from 5.69 to 5.12 with REPA.
>
> > Could you also provide an aligned latent size for better comparison?
>
> Thanks for this great suggestion. We were using 128 tokens mainly for the efficiency consideration, especially for 512x512 generation with SiT-XL.
> We would like to provide a comparison here on the aligned latent size of MAETok with 256 tokens for 256x256 generation with SiT-L in the following:
>
> | Tokens 	| LP 	| rFID 	| gFID 	|
> |:---:|:---:|:---:|:---:|
> | 128 	|  72.3	| 0.48 	|  5.69	|
> | 256 	|  74.5	|  0.37	|  5.05	|
>
> Note that we simply set the learnable token length as 256 for the aligned latent size.
> Ablation results on using only the 256 image tokens without the learnable tokens can be found in our response to Reviewer cxaR.
> We will add these results in our revised paper.
>
> > The authors should discuss with 'Rethinking the Objectives of Vector-Quantized Tokenizers for Image Synthesis'.
>
>
> Thank you for bringing up this relevant paper, and we will include it in our reference with proper discussion in our revised version.
>
> ---
>
> Thanks for the great suggestions and questions again.
> Please let us know if there is further concern.

---

### Official Review · Reviewer_zfeD · 2025-03-14

**Overall Recommendation:** 4

**Summary:**

In this work, the authors find that the latent space with fewer modes and more discriminative features are better for training latent diffusion models. They propose a masked autoencoder for learning the latent space, where the decoder is later finetuned with encoder frozen, and achieve state-of-the-art FID.

**Claims And Evidence:**

Claims are well supported. Both the generation results and visualization of the features support the claims.

**Essential References Not Discussed:**

I did not find missing key related works.

**Experimental Designs Or Analyses:**

Experiments are solid. The authors evaluate on standard imagenet benchmark and have ablation studies on different design choices.

**Methods And Evaluation Criteria:**

The method is consistent with the hypothesis. Evaluation metric is standard.

**Other Comments Or Suggestions:**

Typo L78: as as

**Other Strengths And Weaknesses:**

The findings are interesting and experiments are solid.

This may not be a major weakness, while I wonder that, a more discriminative latent leads a lower FID, would this be related to how FID is computed? If a latent contains more discriminative information, it may also be reflected in the decoded pixel space, which is then used to compuate the FID -- on the features of a deep neural network, where discriminative images may have more stable features. Besides FID, is faster convergence / better quality also observed from visual quality and human evaluation?

**Questions For Authors:**

L134 "the generation quality of diffusion models is dominated by the denoising network’s training loss", does it mean smaller diffusion loss indicates better quality? Based on score matching, I think this may not be always true, especially for different latent spaces?

**Relation To Broader Scientific Literature:**

This work aims at answering the question in the literature -- "what is a good latent space for training latent diffusion model", with a masked autoencoder as the proposed method and state-of-the-art performance.

**Theoretical Claims:**

Theoretical proofs look good to me.

---

> ### Author Rebuttal · Authors · 2025-04-01
>
> Thanks for your suggestions on reviewing our paper. We now address the questions raised as follows.
>
> ---
>
>
> > While I wonder that, a more discriminative latent leads a lower FID, would this be related to how FID is computed? If a latent contains more discriminative information, it may also be reflected in the decoded pixel space, which is then used to compute the FID -- on the features of a deep neural network, where discriminative images may have more stable features.
>
>
> Thanks for your question. A lower FID is largely independent of stable features, as FID reflects the comparison between the mean and covariance of the real and generated data distributions. On the other hand, more discriminative latents describe whether the target latent distributions are separable and are largely independent of the comparison of their mean and variance.
>
>
>
> > Besides FID, is faster convergence / better quality also observed from visual quality and human evaluation?
>
> Yes. Faster convergence can be observed obviously with better visual quality for the images generated during training.
> We list some visualization examples during the training of SiT-L on VAVAE and MAETok at this [Figure link](https://anonymous.4open.science/api/repo/Rebuttal-8656/file/ID14576_maetok_vavae.pdf?v=b6ade62e).
> These visualizations are generated using 30 inference steps and a guidance scale of 1.5 during training.
>
> Also we conducted a human evaluation on 100 images generated by SiT-L trained on VAVAE and MAETok. The win, tie, lose rate of MAETok over VAVAE is shown below:
> |  	| Win 	| Tie 	| Lose 	|
> |:---:|:---:|---|---|
> | MAETok over VAVAE	| 0.75 	| 0.18 	| 0.11 	|
>
> We will add the visualization and human evaluation results into our revised Appendix.
>
> > Typo
>
> Thanks for pointing this out. We will fix this typo in our revised paper.
>
>
> > L134 "the generation quality of diffusion models is dominated by the denoising network’s training loss", does it mean smaller diffusion loss indicates better quality? Based on score matching, I think this may not be always true, especially for different latent spaces?
>
> Sorry for the confusion caused. We found this sentence is indeed not very accurate. What we are trying to convey is that the quality of the learned distribution of diffusion models is dominated by its training loss.
> The eventual generation quality will determined by both the quality of the learned distribution (from the tokenizer encoder) and the capacity of the tokenizer decoder.
> In our theoretical analysis, we assume the tokenizer decoders present similar capacity, and in that case, the generation quality will be determined by the learned distribution and thus diffusion loss. We will fix this sentence in our revised paper to reduce confusion.
>
>
> ---
>
> We hope the above response resolved the questions.

---

### Official Review · Reviewer_pSm9 · 2025-03-16

**Overall Recommendation:** 4

**Summary:**

This paper studies the properties of latent space for diffusion models, and claims that a more discriminative latent space (fewer Gaussian mixture modes) enable more effective diffusion learning and generation quality. Specifically:
1. The paper conducts both empirical and theoretical analysis to show that the latent space with fewer gaussian modes, and thus better separability achieves lower diffusion loss in training.
2. Based on 1, the paper further proposes MAETok, a mask autoencoder architecture which facilitates discriminative feature learning, with the additional learning objectives of predicting semantically rich features like HOG, DinoV2, and CLIP features.
3. Extensive experimental results show that the proposed MAETok achieves state of the art generation results with fewer latent tokens, and thus less computation load, demonstrating the efficacy and validity of the proposed idea.

**Claims And Evidence:**

The claim that discriminative latent space enables effective generative learning is supported by clear and convincing empirical and theoretical analysis.
1. The paper shows that among the different latent spaces learnt by different variations of auto encoders, the ones with less gaussian mixture modes, and thus better separability and discriminative achieve lower diffusion training loss
2. The paper further shows theoretically that with all other conditions kept the same, the latent space that have more gaussian mixture modes require more data samples to achieve the same diffusion loss, further supporting that discriminative latent space facilitates effective diffusion learning with less data samples.
3. Experimental results show the consistency between high linear probing accuracy in latent space (better discriminative features) and better generative quality (through gFID)

**Essential References Not Discussed:**

I haven't found missing essential references.

**Experimental Designs Or Analyses:**

Yes, I checked all the subsections of the Experiment section. Specifically, in section 4.5, the paper points out that with CFG, the proposed method shows worse gFID comparing to previous methods, and hypothesize that the reason is that MAETok already learns a semantically rich latent space, and that the linear scheme of CFG on top of that may not be effective, which is also backed by the tuning results in Appendix C.2. I have a question based on this observation: if learning a discriminative latent space hurts its adaptability with CFG, then how to justify the necessity of MAETok?

**Methods And Evaluation Criteria:**

Yes,  the proposed MAETok adopts a set of training designs to facilitate discriminative feature learning (mask auto encoding, predicting HOG, CLIP, DINOV2); For evaluation, the paper uses linear probing in the latent space to measure discriminability in latent feature space, and also show the corresponding generative performance through gFID, demonstrating the better feature space discriminability indeed facilitate better generation quality.

**Other Comments Or Suggestions:**

please see questions.

**Other Strengths And Weaknesses:**

1. The paper is well written and easy to follow
2. Empirical and theoretical evidence are comprehensive and strongly support the claims

**Questions For Authors:**

1. As mentioned in Experimental design and analysis section, based on the results shown at Table 2, CFG without MAETok shows better gFID than CFG with MAETok. I have two questions regarding this observation and the analysis in the paper.
(1) Why would it be more difficult to apply linear CFG scheme on semantically rich latent features?
(2) If learning a discriminative, semantically rich latent feature space means that it will be more difficult to adapt with CFG scheme, how to justify and value and necessity of MAETok under this setting? Since currently CFG with vanilla VAE features still achieve better results.

**Relation To Broader Scientific Literature:**

To facilitate discriminative latent feature learning, the paper adopts prior ideas like mask autoencoding [1], and further use auxiliary decoders to reconstruct targets such as HOG feature [2], DINOv2 feature [3], and CLIP feature [4], which are known to have good discriminatability.

[1] He, Kaiming, et al. "Masked autoencoders are scalable vision learners." Proceedings of the IEEE/CVF conference on computer vision and pattern recognition. 2022.
[2] Dalal, Navneet, and Bill Triggs. "Histograms of oriented gradients for human detection." 2005 IEEE computer society conference on computer vision and pattern recognition (CVPR'05). Vol. 1. Ieee, 2005.
[3] Oquab, Maxime, et al. "Dinov2: Learning robust visual features without supervision." arXiv preprint arXiv:2304.07193 (2023).
[4] Radford, Alec, et al. "Learning transferable visual models from natural language supervision." International conference on machine learning. PmLR, 2021.

**Theoretical Claims:**

I have looked through the proofs about theorem 2.1 in the supplementary, but I don't think I have strong enough theoretical background to identify any issues.

---

> ### Author Rebuttal · Authors · 2025-04-01
>
> Thanks for your time and efforts reviewing our paper. We now address the raised questions as follows.
>
> ----
>
> > Why would it be more difficult to apply linear CFG scheme on semantically rich latent features?
>
> Thanks for this interesting question.
>
>
> We believe the limitation here is to apply the linear CFG scheme between a **semantic unconditional model** and a **semantic conditional model**.
> There could be two possible reasons why it is more difficult to apply CFG in this scenario.
> The first the linear interpolation could be more often lie outside the latent distribution. where the tokenizer decoder cannot reconstruct/decode reasonably (Fig. 4).
> And the second is the guidance scale becomes more difficult to tune with an unconditional model that already learns a good/semantic distribution (Tab. 5) [1]. In this case, applying CFG to obtain a 'lower-temperature' distribution becomes harder since the original assumption of CFG requires the unconditional model to be free of condition (Eq. (30) in [2]’s Appendix H): $ \hat{q}  ( x_{t+1}  | x_t , c)   \triangleq q(x_{t+1} | x_t )$.
> However, this assumption does not always hold [3], especially in our case.
>
>
> > If learning a discriminative, semantically rich latent feature space means that it will be more difficult to adapt with CFG scheme, how to justify and value and necessity of MAETok under this setting? Since currently CFG with vanilla VAE features still achieve better results.
>
>
> Thanks for this great question.
>
> To justify the value and necessity of MAETok, we need to highlight:
> * **MAETok achieves better results than vanilla CFG with vanilla VAE**. MAETok utlizes an pure autoencoder architecture with **only 128 tokens**. It achieves comparable performance as previous models on 256x256 generation and better results on 512x512 generation with a **vanilla SiT-XL model**, where previous best results are achieved with more training techniques on diffusion models (autoregressive, masking, noise scheduling, etc.). MAETok can also be used with these diffusion training techniques to obtain even better performance.
> * **MAETok works better with more advanced CFG schemes**. The difficulty of adapting naive CFG with MAETok lies in the problem of vanilla CFG, as we discussed before. CFG itself is very tricky and need to be tuned to find the optimal guidance scale. Adopting more recent and advanced CFG schemes can help MAETok achieve better results, as shown below.
>
> | CFG Scheme 	| FID 	| IS 	|
> |:---:|:---:|:---:|
> | Vanilla 	| 1.73 	| 308.4 	|
> | + Bad Version [1] 	| 1.54 	| 315.9 	|
> | + GFT [4] 	| 1.51 	|  312.5	|
>
> These results are obtained using MAETok with SiT-XL on 256 generation. For the bad version guidance [1], we utilized the same SiT-XL model but from 1/4 training (1M steps). Designing more proper bad version models could further improve the results as in [1].
>
> We will add these results in our revised paper.
>
> [1] Karras et al. Guiding a Diffusion Model with a Bad Version of Itself.
> [2] Dhariwal et al. Diffusion Models Beat GANs on Image Synthesis.
> [3] Zhao et al. Studying Classifier(-Free) Guidance From a Classifier-Centric Perspective.
> [4] Chen et al. Visual Generation Without Guidance.
>
> ----
>
> We hope the above response could resolve your questions and if there is further concern please let us know.

---

### Decision · Program_Chairs · 2025-05-01

**Decision:**

Accept (spotlight poster)

**Comment:**

All of reviewers vote for acceptance post-rebuttal. The AC checked all the materials and concurs that the paper has proposed an efficient and effective approach based on masked auto-encoding to train visual tokenizers for diffusion models. The work has made solid contributions to the properties of the latent space that encourages image generation quality, both theoretically and empirically, and therefore should be accepted. Please incorporate necessary changes in the final version, and open-source code/models as promised.